# A generalizable Cas9/sgRNA prediction model using machine transfer learning with small high-quality datasets

Dalton T. Ham[1,3], Tyler S. Browne[1,3], Pooja N. Banglorewala[1], Tyler L. Wilson[2], Richard K. Michael[2], Gregory B. Gloor [1]✉ & David R. Edgell [1]✉

The CRISPR/Cas9 nuclease from *Streptococcus pyogenes* (SpCas9) can be used with single guide RNAs (sgRNAs) as a sequence-specific antimicrobial agent and as a genome-engineering tool. However, current bacterial sgRNA activity models struggle with accurate predictions and do not generalize well, possibly because the underlying datasets used to train the models do not accurately measure SpCas9/sgRNA activity and cannot distinguish on-target cleavage from toxicity. Here, we solve this problem by using a two-plasmid positive selection system to generate high-quality data that more accurately reports on SpCas9/sgRNA cleavage and that separates activity from toxicity. We develop a machine learning architecture (crisprHAL) that can be trained on existing datasets, that shows marked improvements in sgRNA activity prediction accuracy when transfer learning is used with small amounts of high-quality data, and that can generalize predictions to different bacteria. The crisprHAL model recapitulates known SpCas9/sgRNA-target DNA interactions and provides a pathway to a generalizable sgRNA bacterial activity prediction tool that will enable accurate antimicrobial and genome engineering applications.

The Cas9 nucleases from the type II-A clustered regularly interspaced short palindromic repeat (CRISPR) system have gene-editing applications in both bacteria and eukaryotes[1,2]. Cas9 cleavage of DNA templates requires an associated CRISPR RNA (crRNA) that is complementary to the target site, and a trans-activating CRISPR RNA (tracrRNA) that is required for crRNA assembly with Cas9[3]; in most applications these two RNAs are genetically fused into a single guide RNA (sgRNA)[4]. In bacteria, Cas9 nucleases can be used as sequence-specific antimicrobial agents to target distinct bacterial species for elimination[5–12] because many bacteria lack appropriate DNA repair pathways to repair double-strand breaks (DSB). Cleavage by Cas9 causes replication fork collapse and cell death[13]. Alternatively, Cas9 cleavage can eliminate plasmids through the cellular RecBCD exonuclease pathway that degrades linearized DNA. Cas9 can also be used for bacterial genome engineering[14–16], or for transcriptional modulation with catalytically inactive dCas9 variants[17–19].

A major unsolved problem when using Cas9 is the inability to accurately select sgRNA/ Cas9 combinations that lead to high on-target activity in both eukaryotic and prokaryotic systems. Selection of sgRNAs typically involves computational prediction of activity where the underlying models are trained on data of in vitro or in vivo Cas9/sgRNA activity, and may also include biochemical parameters of Cas9 activity, biophysical calculations of sgRNA:DNA stability, and chromatin accessibility information[20–25]. However, as recently reported[26,27], most computational models poorly predict sgRNA activity outside of the dataset on which they are trained. This lack of generalizability could be because the underlying data are sparse and not independently validated, because the datasets may not accurately represent Cas9/sgRNA cleavage activity and instead report a secondary DNA repair outcome of DSB generation, because the machine learning algorithms are not optimal, or a combination of all three[26].

---

[1]Department of Biochemistry, Schulich School of Medicine and Dentistry, London, ON N6A5C1, Canada. [2]Tesseraqt Optimization Inc, Toronto, ON, Canada. [3]These authors contributed equally: Dalton T. Ham, Tyler S. Browne. ✉e-mail: ggloor@uwo.ca; dedgell@uwo.ca

In spite of the conceptual simplicity in targeting sgRNAs to small bacterial genomes, eukaryotic-based computational models fail to accurately predict activity in bacteria[28]. One issue for sgRNA activity predictions in bacteria is that there are few bacteria-specific large-scale datasets of Cas9/sgRNA activity[29,30]. In each case, deep sequencing was used to readout sgRNA abundance of a pooled sgRNA library targeting the *Escherichia coli* genome, with the assumption that sgRNA depletion was correlated with active Cas9/ sgRNA combinations. A complicating factor in assessing Cas9/sgRNA activity in bacteria is that expression of Cas9 (and dCas9) alone can result in cellular toxicity and slow growth[31–34]. Thus, experimental strategies that only use bacterial killing as a measure of Cas9/sgRNA activity cannot separate toxicity from activity because both will result in depletion of sgRNAs from a pooled high-throughput experiment. Two sgRNA prediction models have been developed based on this data, sgRNA-cleavage-activity-prediction[29] and DeepSgRNAbacteria[35], but we found poor correlation between predicted Cas9/sgRNA activity and killing of *Salmonella typhimurium*[5]. Other factors that possibly impact sgRNA activity in bacteria include sub-optimal secondary structures in the crRNA and tracrRNA[36], and similarity between the crRNA seed region and so-called "non-targets" in bacterial genomes. In contrast, DNA modifications do not impact activity of type II CRISPR systems (from which Cas9 is derived)[37,38]. Similarly, there is no bias in activity for Cas9/sgRNAs targeting the template or non-template strand of transcribed genes, or in targeting the leading or lagging strands relative to DNA replication origins[5].

Taken together, the evidence indicates that there is a pressing need for additional high-quality bacterial sgRNA activity data sets to validate and generalize previous findings, and to provide training data for predictive machine learning models. Here, we develop a paired experimental design in *E. coli* that compares behaviors of Cas9/sgRNA combinations in repressed and induced conditions to provide a readout of activity where active sgRNAs are enriched in a pooled library. This approach differs from previous depletion studies by accounting for initial sgRNA abundance in the pooled library, and does not rely on end-of-experiment sgRNA abundance as the sole indicator of sgRNA activity. Additionally, this setup distinguishes highly active Cas9/sgRNA combinations from toxic ones with poor growth, even in repressed conditions. We use this approach with the SpCas9 nuclease[4] and the TevSpCas9 dual-nuclease[39] to generate robust sgRNA activity datasets to train a sgRNA prediction model, crisprHAL (crispr macHine trAnsfer Learning) that recapitulates the known biology of the Cas9/sgRNA-target DNA interaction surface. Significantly, we find that transfer learning from existing datasets with a small amount of sgRNA activity data (279 sgRNAs) from our assays improved bacterial sgRNA predictions relative to previous models. Crucially, crisprHAL

can generalize Cas9/sgRNA activity predictions to different bacteria. Collectively, our study highlights the importance of accurate sgRNA activity data and transfer learning as being critical for computational modeling.

## Results

### Current bacterial sgRNA prediction models are poorly generalizable

We were interested in understanding why existing sgRNA prediction models[29,35] poorly correlate with in vivo activity[5]. Thus, we tested whether current bacterial sgRNA prediction models were generalizable to different SpCas9 activity datasets (Fig. 1). For this, we used a two-plasmid positive selection system (Fig. 2) to generate two high-quality activity datasets for the SpCas9 and the TevSpCas9 dual nuclease (as described in detail in the following sections). When the TevSpCas9 dataset was used as an input for the sgRNA-cleavage-activity-prediction model[29] (hereafter referred to as the Guo model) and the DeepSgRNAbacteria model[35] (hereafter referred to as the DeepSgRNA model), we found only modest predictive performance between predicted activity and experimental results, as measured by Spearman correlation of rank order (Fig. 1). Modest predictive power was observed regardless of which of the two published sgRNA depletion datasets the Guo or DeepSgRNA models were trained on; one dataset used SpCas9 and the other used an enhanced high-fidelity SpCas9 variant (eSpCas9). We also tested whether 4 eukaryotic sgRNA prediction models (DeepHF[40], C-RNNCrispr[41], DeepSpCas9[42], Crispr-NET[43]) were generalizable to the SpCas9 and TevSpCas9 activity datasets and found Spearman ranked correlation coefficients of between −0.2 and 0.1 (Supplementary Fig. S1). DeepGuide is a recently developed sgRNA activity prediction model for the yeast *Yarrowia lipolytica*[44] that was suggested as being applicable for bacterial sgRNA predictions. We retrained the DeepGuide model with the Guo eSpCas9 dataset and then tested predictions using the TevSpCas9/sgRNA activity data generated here; we found a rank correlation of 0.505 between predicted and measured activity. These results emphasize a major issue with Cas9/sgRNA activity predictions, namely the lack of generalizability and accuracy when models are used with data outside of the initial training data and that existing eukaryote-specific models datasets cannot predict activity when trained with bacterially-derived Cas9 datasets. Collectively, these observations highlight the need for high-quality datasets that accurately report on Cas9/sgRNA cleavage activity.

### Profiling sgRNA activity using a two-plasmid system

To increase the accuracy of SpCas9 and TevSpCas9 targeting predictions, we started with an improved assay in which we used an

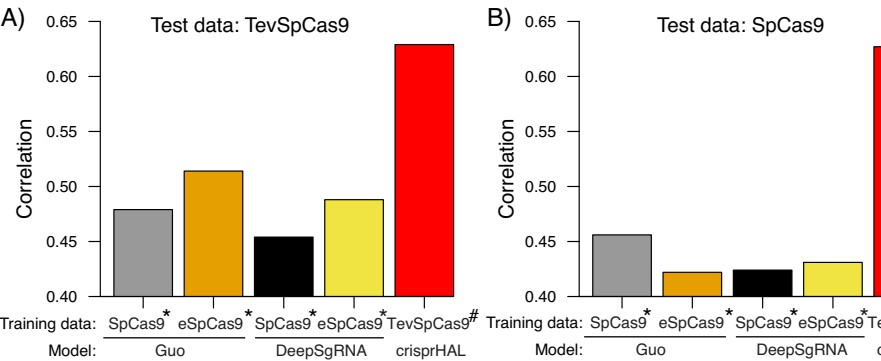

**Fig. 1 | Spearman ranked correlation of predicted versus measured activity for sgRNA prediction models.** Barcharts are Spearman Rank correlations between the (**A**) TevSpCas9 dataset (n = 279) and (**B**) the SpCas9 dataset (n = 303) generated in this study and predictions from bacterial sgRNA activity models including crisprHAL. The crisprHAL values are reported as the average rank correlation from 5-fold cross validation. For both panels, asterisks (*) indicate datasets from ref. 29 and hash marks (#) indicate datasets generated in this study. Source data are provided as a Source Data file.

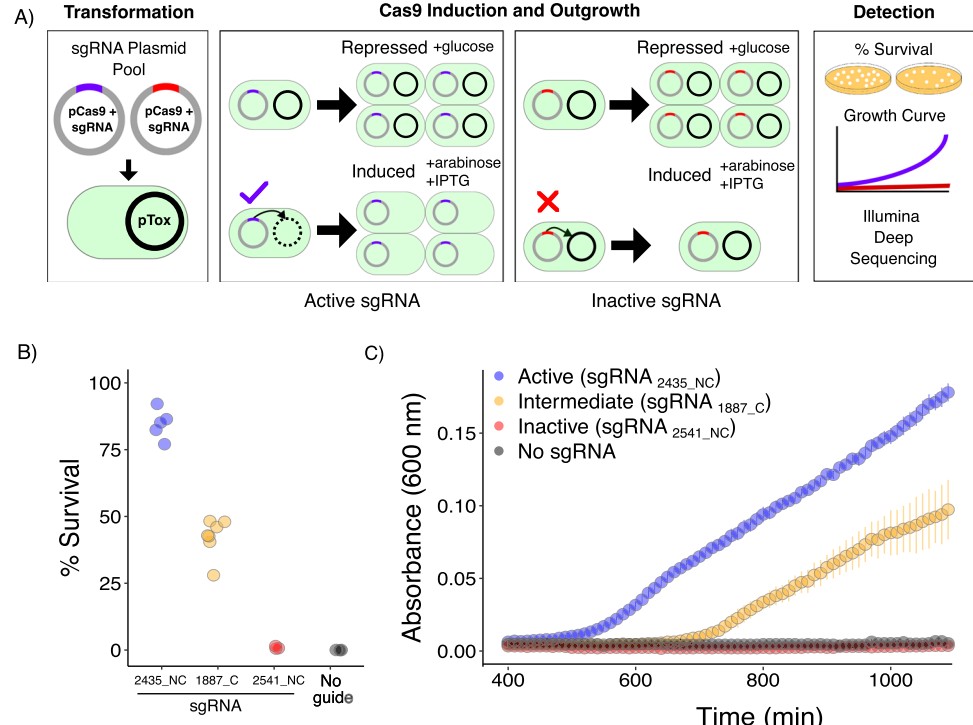

**Fig. 2 | Two-plasmid survival assay. A** Experimental workflow of the two-plasmid system. *Transformation*, the pCas9 plasmid expressing SpCas9 or TevSpCas9 from an arabinose-inducible promoter and a sgRNA from a constitutive tetracycline resistance gene promoter is transformed into *E. coli* harboring pTox. *Induction and Outgrowth*, transformed cells are split into repressed (0.2% D-glucose) or induced (0.02% L-arabinose and 0.4 mM IPTG) conditions and grown for 18 h. *Active sgRNAs, blue* promote robust cleavage of the toxic plasmid and cell growth while *inactive sgRNAs, red* do not cleave pTox preventing cell growth. *Detection*, SpCas9/sgRNA activity can be read out by (i) deep-sequencing the pCas9 sgRNA cassette, (ii) growth curves that measure optical density of induced and repressed cultures, or (iii) plating on solid media to determine a percent survival based on the ratio of colonies on induced media (chloramphenicol and IPTG) and repressed media (chloramphenicol and D-glucose). **B** Different TevSpCas9/sgRNA combinations promote a range of survival. Plot of survival percentage for three different sgRNAs targeted to pTox (2435_NC,1887_C,2541_NC) identified as active (blue), intermediate (orange), inactive (red) as well as a no-sgRNA(NG) control (black). Individual data points represent independent experiments. **C** Growth curve of *E. coli* harboring the SpCas9/sgRNA combinations used in (**B**) plotted as time versus absorbance at 600 nm. Data points represent the mean of three biological replicates and the whiskers representing the standard deviation from the mean. Source data are provided as a Source Data file.

integrated approach to assess SpCas9/sgRNA activity in *E. coli* (Fig. 2A). We adapted a two-plasmid system used for in vivo selection experiments[45–47] that is known to correlate with enzymatic activity in vitro[48] and expressed the SpCas9 or dual-nuclease TevSpCas9 protein (arabinose inducible) and a sgRNA (constitutive expression) from one plasmid (pSpCas9/ or pTevSpCas9, Supplementary Fig. S2) in combination with a second plasmid (pTox, Supplementary Fig. S2) harboring the *ccdB* DNA gyrase toxin controlled by an IPTG inducible *lac* promoter (Fig. 2A). Cleavage of the pTox plasmid by an active SpCas9/sgRNA combination or TevSpCas9/sgRNA combination (Fig. 2B) leads to degradation of the pTox plasmid and subsequent cell growth and enrichment of cells lacking the pTox plasmid in the population. Inactive SpCas9/sgRNA or TevSpCas9/sgRNA combinations do not eliminate the pTox plasmid and are unable to grow under toxin-inducing conditions. Importantly, the activity of the (Tev)SpCas9/sgRNA combination is related to the rate of pTox plasmid clearance, and so partially active combinations will have intermediate outgrowth and lethality characteristics. With this system, (Tev)SpCas9/sgRNA activity can be analyzed by deep sequencing of the sgRNA expression cassette following competitive growth in liquid media, or by growth rate in liquid media, or by counting colonies grown on solid media (Fig. 2A). The dual-active-side nuclease TevSpCas9 has an extended targeting requirement that includes the 5′-CNNNG-3′ I-TevI cleavage motif (Supplementary Fig. S3)[39]. Thus, all TevSpCas9 sites are also SpCas9 sites, and cleavage by an active TevSpCas9/sgRNA combination will create an additional DSB with the potential to enhance killing efficiency.

We validated this system by targeting three TevSpCas9/sgRNA combinations to a unique region of pTox; sgRNA_{2435_NC}, sgRNA_{1887_C}, and sgRNA_{2541_NC} (in this naming scheme sgRNAs are identified by the position of the PAM-distal nucleotide of the sgRNA target in pTox and whether they target the coding or non-coding strand, as all genes are in the same orientation). We plated the transformed *E. coli* cells on solid media and calculated percent survival by comparing the proportion of colony forming units on toxin-inducing or toxin-repressing agar plates. When expressed in combination with the TevSpCas9 protein, the three sgRNAs tested showed survival ranging from 88.2 ± 4.1% (standard error of the mean) for sgRNA_{2435_NC} to 0.9 ± 0.29% for sgRNA_{2541_NC} (Fig. 2B). When no sgRNA was present (NG, no guide), we observed 0% survival (Fig. 2B). We conducted a similar experiment in liquid media by measuring absorbance at 600 nm over 18 h to detect growth under inducing and non-inducing SpCas9 conditions in combination with the same three sgRNAs (Fig. 2C). The resulting growth curves are consistent with the survival values on solid media, with sgRNA_{2435_NC} promoting robust growth, sgRNA_{1887_C} promoting intermediate growth and sgRNA_{2541_NC} and the NG control showing no growth (Fig. 2C). Collectively, these results show that bacterial growth is dependent on cleavage of the pTox plasmid by TevSpCas9/sgRNA, agreeing with previous results using SpCas9[47], and that differential TevSpCas9/sgRNA activity results in distinct growth differences over a large and consistent range.

### Sensitivity of the two-plasmid system
We next tested the ability of the two-plasmid system to detect changes in SpCas9/sgRNA or TevSpCas9/sgRNA activity when read out via a

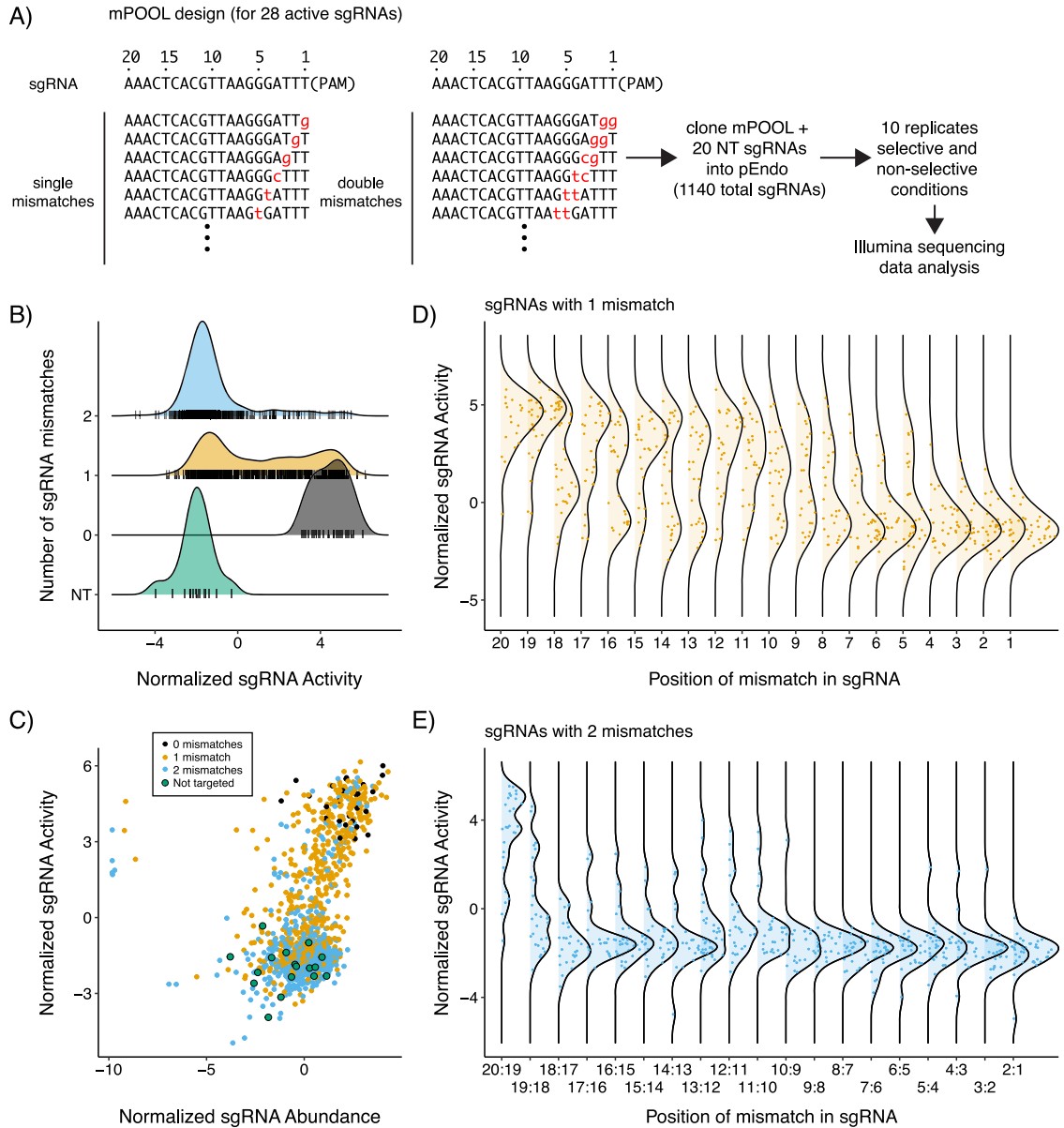

**Fig. 3 | Activity of sgRNAs with single and double mismatches. A** Schematic of the mutant pool (mPool) design and experimental approach. Single and dinucleotide transversions are indicated by lower case red letters, with sgRNAs numbered from PAM proximal (postion 1) to PAM distal (position 20). **B** Ridge plots of normalized sgRNA activity scores for non-targeted sgRNAs (NT, green) perfectly matching sgRNAs (black), sgRNAS with single nucleotide mismatches sgRNAs (yellow), and sgRNAs with dinucleotde mismatches (cyan). **C** Bland-Altmann plot comparing the normalized abundance and normalalized activity scores for sgRNAs in the mPool with the colors representing the same sgRNAs categories as (**B**). Ridge plots of normalized sgRNA activity scores by position of mismatch for sgRNAs with (**D**) single or (**E**) 2 mismatches. Source data are provided as a Source Data file.

multiplexed high-throughput sequencing experiment. This experiment was designed to validate the sensitivity of the two-plasmid system when reporting on a range of TevSpCas9/sgRNA activities, and to assess the effect of mismatches between the sgRNAs relative to their cognate target site. For this, we designed an oliognucleotide pool where single and double nucleotide transversions were tiled along the length of 28 different sgRNAs that were targeted to a unique 3.2 kb region of the pTox plasmid (Fig. 3A, Supplementary Data 1). The mutated oligonucleotide pool (mPool) also contained 20 sgRNAs not targeted to pTox and 28 exactly matching sgRNAs as internal controls, for a total of 1140 sgRNAs. The mPool was cloned into pTevSpCas9 and we performed 10 independent transformations into *E. coli* harboring pTox. Each transformation culture was split and then grown under conditions that repressed or induced TevSpCas9 and CcdB. We anticipated that active TevSpCas9/sgRNA combinations would

become enriched under the inducing conditions relative to the pool grown under non-induced conditions. Our output score (reported as normalized sgRNA activity) was the log2 difference in relative sgRNA abundance between the induced and uninduced conditions (as described in the Methods). Given the solid and liquid culture results (Fig. 2), we anticipated that the assay would report a distribution of activities that depended on the underlying activity of the pTevSpCas9/sgRNA combination. After Illumina sequencing of the sgRNA cassette from both conditions and data analyses, active combinations were identified by a higher normalized activity score (Supplementary Data 2).

As expected, when co-expressed with TevSpCas9, sgRNAs that exactly matched their target sequences (black) (Fig. 3B, C) tended to exhibit high normalized activity scores, sgRNAs with single mismatches to their target site (orange) showed a broad range of

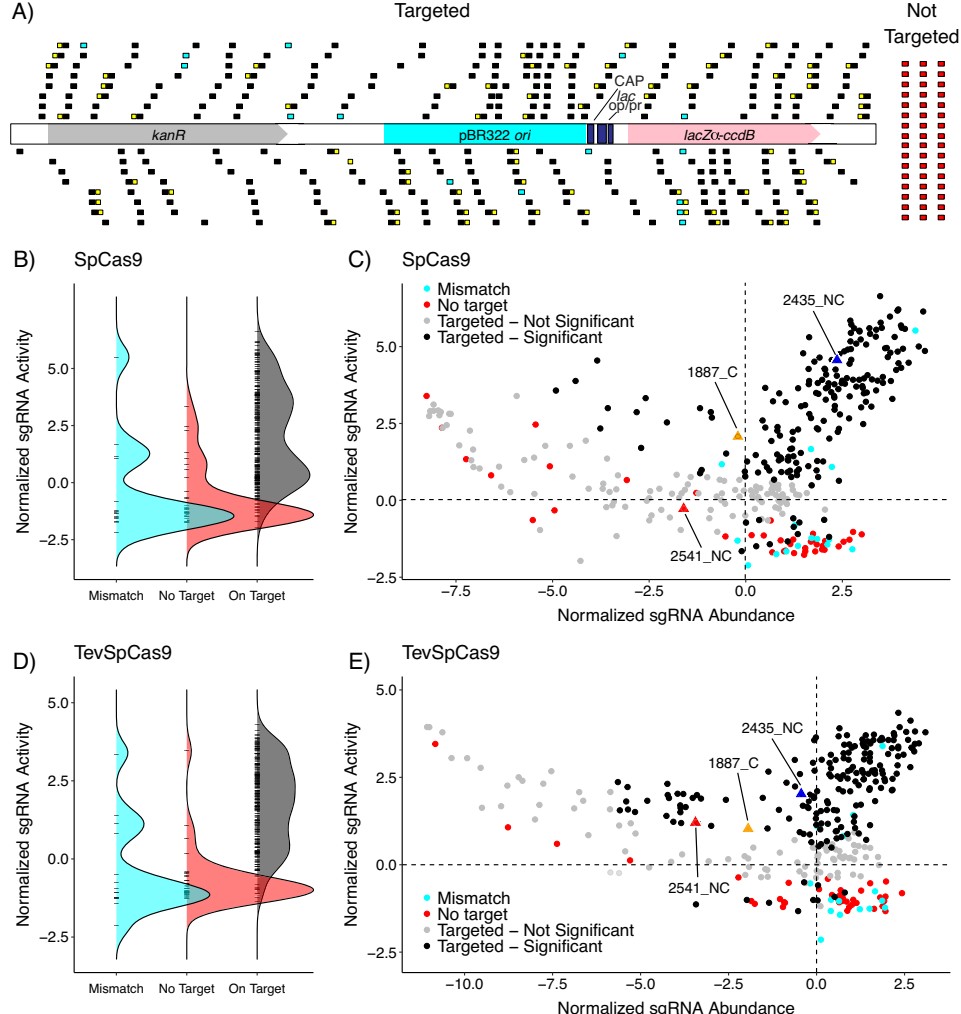

**Fig. 4 | High-throughput pooled screen detects distribution of SpCas9/sgRNA and TevSpCas9/sgRNAs activity.** **A** Schematic of target sites for the sgRNAs pPool with black boxes representing sgRNA target site (304), cyan boxes representing target sites with mismatches (15), red boxes representing non-targeting sgRNAs (48) and yellow boxes representing TevCas9 sites (75). Distribution of normalized activity scores for mismatched (cyan), non targeting (red), and on target (black) sgRNAs for Cas9 (**B**) and TevCas9 (**D**) experiments. Bland-Altmann plots comparing the normalized abundance and activity scores for individual sgRNAs in the Cas9 (**C**) and TevCas9 (**E**) pooled experiments. sgRNAs with a false-discovery rate (FDR) <0.01 are highlighted black and sgRNAs with a FDR >0.01 are colored gray. Cyan and red points represent mismatched sgRNAs and non-targeting sgRNAs respectively. sgRNAs that were tested individually in Fig. 2B, C are shown as triangles where 2435_NC is blue, 1887_C is orange and 2541_NC is red. Source data are provided as a Source Data file.

activities, and sgRNAs with double mismatches to their target site (blue) generally had low activity scores that were similar to non-targeted sgRNAs (green). Also as expected, the ability of the sgRNA to confer activity was most impacted by mismatches in the seed region corresponding to positions 1–10 relative to the PAM proximal end (Fig. 3D)[49,50]. The impact of double transversions was more pronounced than that of single transversions. In the former, mismatches in all positions except 20 and 19 severely reduced activity (Fig. 3E), while in the latter there was a broader range of activity conferred (Fig. 3D). These results agree with previous studies on mismatch tolerance of Cas9/sgRNA from in vitro data and eukaryotic systems[51–53] emphasize that selection of appropriate sgRNAs without mismatches is critical for bacterial applications where specificity is a concern[8]. The data also show that our experimental system can report a gradient of sgRNA activities across an ~1000-fold normalized activity range and a ~2000-fold range in relative abundance; although the relative abundance range was more clustered except for a few outlier sgRNA sequences.

## High-throughput profiling of a pooled sgRNA library

We next synthesized an oligonucleotide pool (oPool) to interrogate the activities of 304 exact match sgRNAs targeted to the pTox plasmid, with all sgRNA sites having a 5′-NGG-3′ PAM sequence (Fig. 4A, Supplementary Data 3). The oPool also contained 15 sgRNAs with nucleotide mismatches that had varying degrees of target complementarity to the pTox plasmid, and 48 sgRNAs that did not have any complementarity to the pTox plasmid. In addition, 73 of the 304 sgRNAs that exactly matched their target sequence also contained an exact match with a consensus I-TevI cleavage site at the correct spacing from the SpCas9 binding site. In total, the oPool contained 367 sgRNAs (Supplementary Data 3). The oPool was cloned into pTevSpCas9 and pSpCas9, and 10 transformation replicates for each was generated. Following induction and outgrowth, the result was read out by Illumina sequencing and data analysis to assign normalized activity and relative abundance scores for each sgRNA in combination with both SpCas9 and TevSpCas9 (Supplementary Data 4). The final dataset included 332 sgRNAs and the major findings from these experiments were:

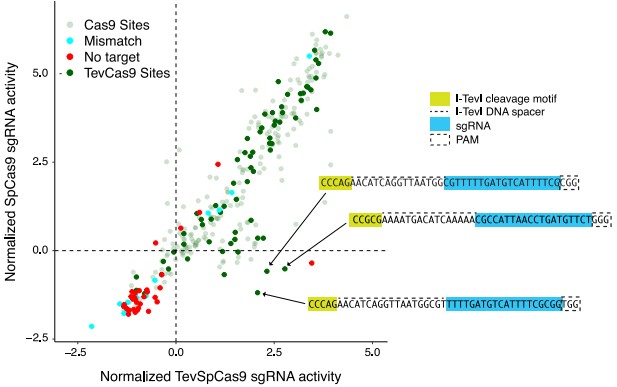

**Fig. 5 | Activity of TevSpCas9 versus SpCas9 with the pooled sgRNA library.**
Comparing the difference between condition values for sgRNAs present in both TevSpCas9 and SpCas9 pooled experiments where dark green dots represent sgRNAs with upstream I-TevI recognition sites and light green dots representing sgRNAs with Cas9 sites only. Non-targeting and mismatched sgRNAs are highlighted as red and cyan respectively. Three sgRNAs that target TevSpCas9 sites are indicated. Source data are provided as a Source Data file.

1. Of the 304 sgRNAs with perfect complementarity to the pTox plasmid, 174 had significant positive normalized activity scores in the SpCas9 data set and 178 in the TevSpCas9 data set using a false-discovery rate <0.01 (Fig. 4C, E).
2. The non-targeted (red) and mismatched (cyan) sgRNAs generally had negative normalized activity scores indicating that they did not cleave the pTox plasmid efficiently (Fig. 4B−E).
3. We found no nucleotide preference in the first position of the 5′-NGG-3′ PAM for either SpCas9 or TevSpCas9 (Supplementary Fig. S4).
4. sgRNA relative abundance alone was misleading as a measure of activity as the vast majority of sgRNA sequences were highly abundant, and both mismatched and non-targeted sgRNA sequences tended to be more abundant than average (Fig. 4C, E).

One interesting finding from the oPool experiment was the activity of sgRNAs in the SpCas9 versus the TevSpCas9 experiment. Overall, the readouts from the same sgRNAs in both assays behaved similarly (Fig. 5, Pearson correlation 0.90, *p*-value < $2.2 \times 10^{-16}$), but we found 22 sgRNAs that promoted higher activity with TevSpCas9 than with SpCas9. In the SpCas9 experiment, these sgRNAs had low normalized activity scores ranging from −1.21 to 1.45 versus −1.01 to 2.77 in the TevSpCas9 experiment. The single non-targeted sgRNA (NT42) with a high activity of 3.4 in the TevSpCas9 experiment also showed high replicate-to-replicate variability suggesting that this was an outlier (Supplementary Data 4). One explanation for the increased activity of sgRNAs in the TevSpCas9 experiment was the presence of the I-TevI 5′-CNNNG-3′ cleavage motif at an appropriate distance upstream of the sgRNA binding site (Figs. 2B and 5). This observation suggests that SpCas9 binding is necessary but not sufficient for cleavage, and that low SpCas9 cleavage can be rescued by the I-TevI nuclease domain to promote elimination of the pTox plasmid.

We also noted a large dynamic range for the normalized activity scores (~1000-fold) and relative abundances of the sgRNA sequences (~2000-fold) (Fig. 4C, E). The dynamic range allowed us to identify sgRNAs with low abundances but large activity scores (upper left quadrant of Fig. 4C, E), and would consider these sgRNAs as potentially toxic. Conversely, we identified sgRNAs with high abundance but negative activity scores (lower right quadrants of Fig. 4C, E); 58.7% and 73.5% of these sgRNAs are non-targeting (red) or mismatched (cyan) guides with respect to the pTox plasmid.

The observation that the final datasets included 332 of the 367 designed oPool sgRNAs suggested that these missing sgRNAs are toxic because they contain sufficient identity to promote cleavage of the

*E. coli* chromosome and thus are unclonable (Supplementary Data 5). We classify these sgRNAs as overtly toxic and they were excluded from the training and test datasets for model development and testing. We further confirmed this observation by identifying sgRNAs that were present in the cloning reaction but missing from the pool of recovered plasmids after transformation into *E.coli* Epi300 (Supplementary Fig. S5). Of the clonable sgRNAs that we identified in the upper left quadrant of Fig. 4C, E two of the sgRNAs are exact matches to the *lac* regulatory region present on pTox and in the E. coli chromosome suggesting that toxicity arises from cleavage of the chromosomal target, but only under inducing experimental conditions (Supplementary Fig. S6). However, the other sgRNAs identified in the upper left quadrant of Fig. 4C, E have 5 to 8 nucleotide mismatches that are inconsistent with off-target cleavage based on our mismatch tolerance profiling (Fig. 3, Supplementary Fig. S6).

Taken together, the data highlight the importance of conducting an experiment where the paired design allows the readout of relative enrichment with multiple replicates to accurately measure the ability of sgRNA to confer activity on the complex. Moreover, the approach demonstrates that using sgRNA relative abundance alone as an indicator of activity can lead to false identification of the abilty of sgRNAs to confer activity.

## Growth curves of individual sgRNAs identifies toxic guides
The pooled sgRNA experiments in Figs. 3 and 4 revealed a wide range of sgRNA activity. To cross validate these activity measurements we blindly picked 77 colonies from the transformed pTevSpCas9/sgRNA-oPool library to test using individual growth experiments as shown in Fig. 2D; the identity of each sgRNA was confirmed by sequencing of isolated plasmids. We rationalized that growth curves performed with individual sgRNAs would better resolve measure the properties of sgRNA species independent of their behavior in a sgRNA pool where we could only measure relative changes. These experiments were performed when the TevSpCas9 protein and the CcdB proteins were induced or repressed, and we found three different classes of sgRNA sequences (Fig. 6A–C, Supplementary Fig. S7). Those sgRNAs that conferred a high level of activity when complexed with TevSpCas9 (20 of 77) grew in both induced and repressed conditions (Fig. 6A) whereas inactive sgRNAs (12 of 77) only grew in the repressed condition (Fig. 6B). Surprisingly, we found a number of sgRNAs that we classified as toxic (12 of 77) because they grew poorly in both the induced and repressed condition (Fig. 6C) as compared to a non-targeting sgRNA (Fig. 6D). The growth curves for the remaining 33 sgRNAs did not clearly fit in any category but showed intermediate activity. For active sgRNAs, we consistently found that maximal optical density values were lower in the induced than the repressed condition. We attribute this difference to the presence of glucose in the media used for the repressed condition, which is a preferred carbon source to the arabinose present in the media for the induced condition.

For each sgRNA, we calculated the area under the curve (AUC) for the induced and repressed conditions and normalized them relative to the average AUC for all sgRNAs for each condition (Fig. 6E, Supplementary Table S1). This plot emphasizes that many guides conferring activity grew well in both induced and repressed conditions (20 of 77, black dots Fig. 6E). Conversely, a subset of sgRNAs showed poor or no growth in induced conditions, but robust growth in repressed conditions, and thus were considered inactive (12 of 77, red dots in Fig. 6E), although there was no clear separation between these two groups. This analysis also revealed that toxic sgRNAs grew poorly in both repressed conditions and induced conditions (12 of 77, cyan dots Fig. 6E). We considered that toxicity could be due to off-target sgRNA sites in the E. coli genome. Using Cas-OFFinder[54] we found that none of these sgRNAs have sites with less than four mismatches making off-target cleavage unlikely (Supplementary Table S2). This suggests that toxicity is either an intrinsic property of the sgRNA or that these sgRNAs confer

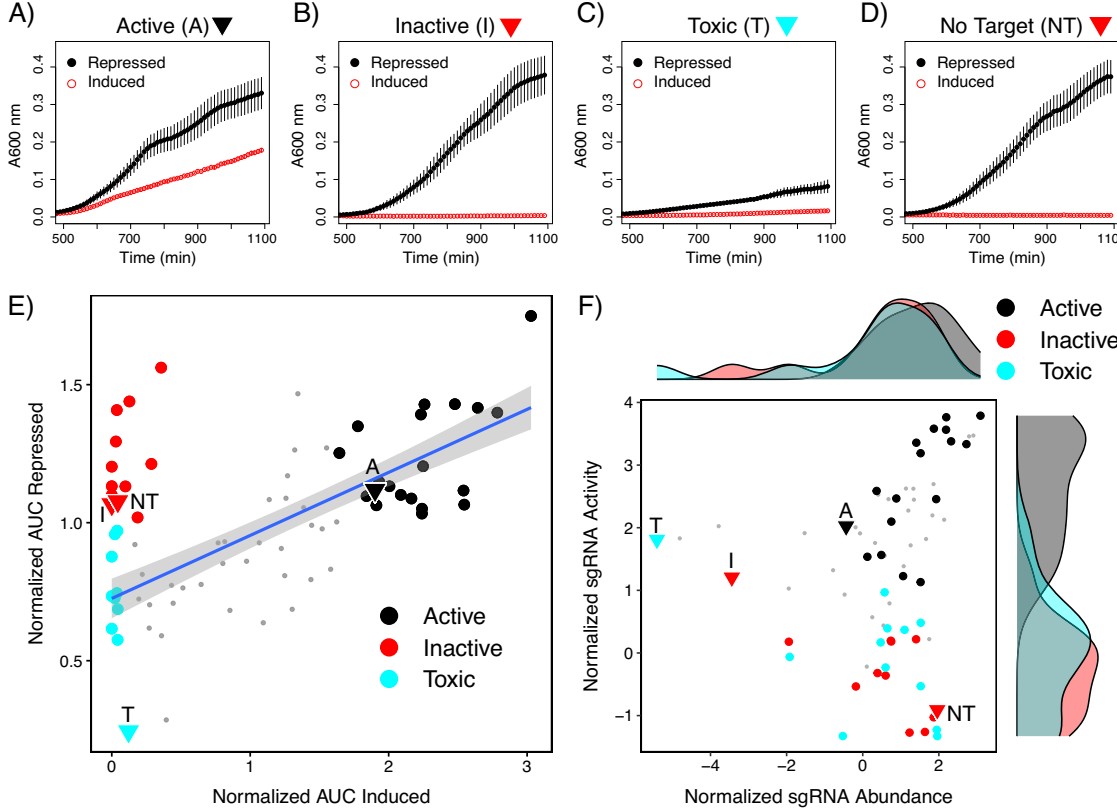

**Fig. 6 | Assaying sgRNAs individually identifies distinct phenotypes.** Representative growth curves of active (**A**), inactive (**B**), toxic (**C**) and non-targeted (**D**) sgRNAs under induced (red dots and line) and repressed (black dots and line) conditions. Points are the mean of three biological replicates and whiskers represent the mean plus or minus the standard deviation. Growth curves for all tested sgRNAs are in Supplementary Fig. S7 and Supplementary Table S1. **E** Plot of AUC for induced and repressed conditions for all sgRNAs. sgRNAs were classified as active (black dots, AUC > 1.64) or toxic (cyan dots, AUC < 0.121) based on quantiles of the AUC for the induced condition. Gray points are sgRNAs with an intermediate phenotype. **F** Marginal density plot of normalized activity and abundance for each sgRNA using data shown in Fig. 4E. Gray points are sgRNAs with an intermediate phenotype. Source data are provided as a Source Data file.

some unwanted property on the TevSpCas9 protein when complexed with the toxic sgRNA.

To address parallels between individual and pooled experiments, we mapped the different classes of sgRNAs from the growth experiments back to the analyses of the deep sequencing experiments (Fig. 6F, G). This revealed that sgRNAs that were classed as inactive in the growth experiments had poor activity in the pooled experiments, with a mean normalized activity score of −0.309 and relative abundance value that were suggestive of minimal or modest enrichment (Fig. 6F). In contrast, sgRNAs conferring activity in the growth experiments largely had positive activity scores and relative abundance values (Fig. 6F). Interestingly, guides determined to be toxic by the growth curves had activity scores ranging from −1.33 to 1.81 in the pooled experiment (6F, mean value of −0.0431) and many of these sgRNAs had positive relative abundance values (Fig. 6F). One explanation for this apparent discrepancy between toxicity and activity is that toxic sgRNAs vary in how they promote bacterial growth in the repressed and induced conditions. For instance, a toxic sgRNA may still be active on the intended pTox plasmid target site under inducing conditions (thus promoting growth), but show toxicity under repressive conditions (thus preventing growth) in turn altering the relative difference calculation that is used to infer activity.

Collectively, these data emphasize the importance of independent validation of sgRNA activity using different methods of activity assessment. Our analyses revealed that many sgRNAs that would be considered active solely by their relative abundance in deep sequencing experiments demonstrated high levels of toxicity when analyzed individually. Thus, toxicity and high activity are not mutually exclusive,

but cannot be distinguished if sgRNAs are classified based on a single line of experimental evidence. Our data also suggest that toxicity is not an all-or-nothing phenotype. In contrast, sgRNAs that are clonable but show toxicity possess mismatches to chromosomal targets at positions that do no support cleavage by our mismatch profiling (Fig. 3). Further studies are needed to directly identify toxic sgRNAs on a large scale to determine the mechanism(s) of toxicity.

### Transfer learning is required for suitable TevSpCas9 predictive ability

With this data in hand, we next concentrated on building a model, crisprHAL (Fig. 7, Supplementary Data 6 and Supplementary Data 7), that could more accurately predict sgRNA target site sequence-associated TevSpCas9 and SpCas9 activity in *E. coli* and other bacteria. For this, we constructed a dual branch deep learning model and tested this model using the TevSpCas9 dataset generated in this study. To select our network architecture and evaluate model performance, we used 5-fold cross validation, measured by Spearman ranked correlation coefficient, hereafter referred to as rank correlation.

Our initial model tests resulted in poor performance, with a rank correlation of 0.308. This result is not surprising given the small size of the TevSpCas9 dataset (*n* = 279) (Fig. 7). Thus, we chose to pursue transfer learning to improve performance. We constructed new models to transfer learn from – denoted as base models – on SpCas9 (*n* = 40308) and eSpCas9 (*n* = 45010) datasets derived from an sgRNA depletion experiment in *E. coli*[29]. Hereafter, these data are referred to as the Guo SpCas9 or Guo eSpCas9 datasets to distinguish them from the SpCas9 dataset generated in this study. A major difference of our

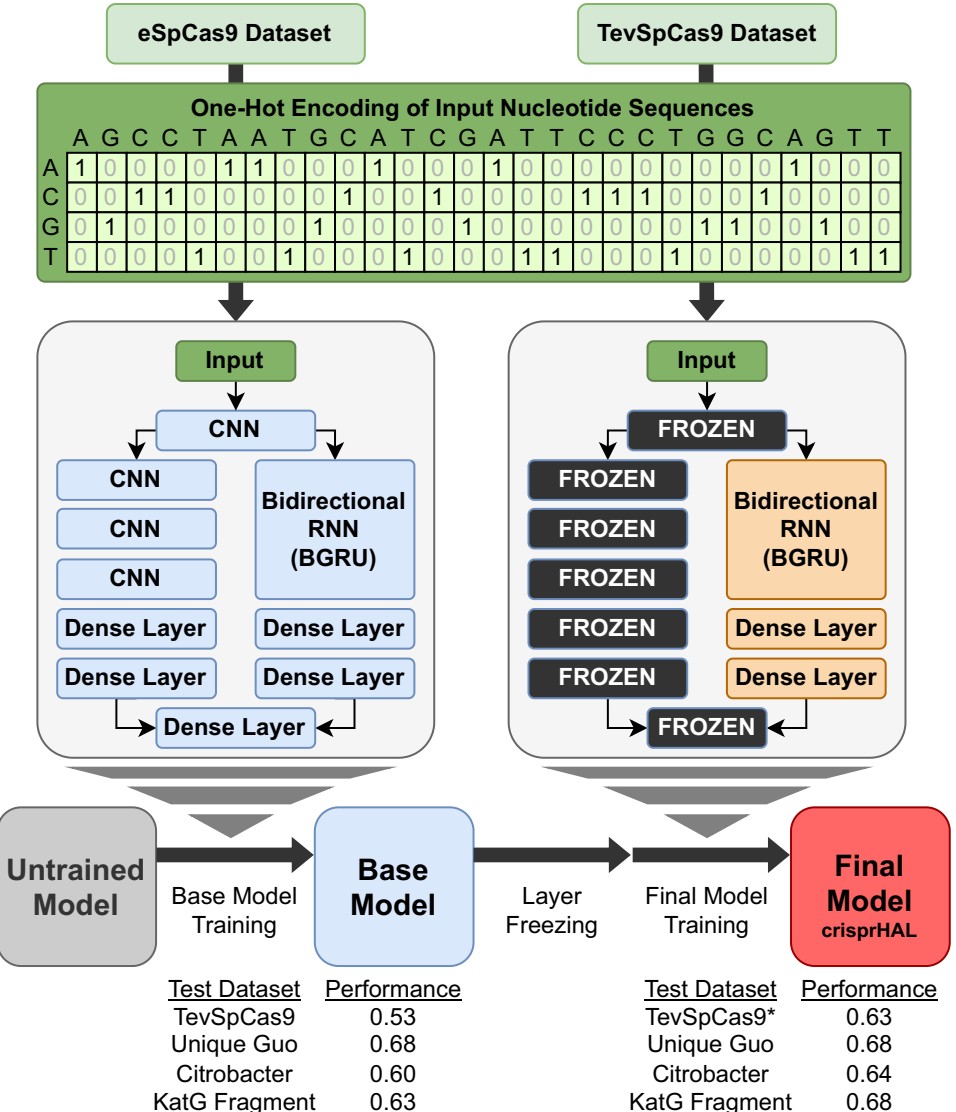

**Fig. 7 | crisprHAL model architecture and transfer learning process.** At top is the schematic showing the one-hot encoding of the input sgRNA sequences from the eSpCas9 or TevSpCas9 datasets used in training the base model (left) and the transfer learning final model (right). The dual branch 4-layer CNN and hybrid CNN-BGRU RNN architectures with base model trainable layers (blue) is on the left versus the final crisprHAL model with trainable (orange) and frozen (black with white text) layers on the right. Performance from the base model and post-transfer learning final crisprHAL model tested on the TevSpCas9 ($n = 279$), unique Guo SpCas9 ($n = 7821$), *Citrobacter rodentium* TevSpCas9 ($n = 30138$), and *S. entericakatG* fragment TevSpCas9 ($n = 228$) datasets with rank correlation scores are displayed at the bottom. The asterisk (*) indicates performance from the final model, crisprHAL, on the TevSpCas9 dataset, as measured by the average rank correlation from 5-fold cross validation.

model compared to prior models is the use of a log ratio-based relative difference metric for scoring sgRNA-associated nuclease activity[55]. This metric resulted in mean scores near 0 across all datasets, while providing differences in their dynamic ranges. The Guo SpCas9 and eSpCas9 datasets contained the widest range of scores, with standard deviations of 2.280 and 2.492, respectively. The TevSpCas9 dataset contained the smallest range, with a standard deviation of 1.305. To compensate for the variations in activity score ranges, we standardized the scores for each dataset by dividing by the standard deviation (Supplementary Fig. S8, Supplementary Data 8). This improved model performance since each dataset was on the same scale. Base model performance was unaltered as expected since this is a simple linear scaling.

Following base model construction, we tested variations in freezing parameter weights for specific layers within the model to optimize for transfer learning performance. We found that freezing the multi-layer CNN branch and leaving all layers of the CNN-BGRU branch

– except for the initial CNN layer shared by both branches of the model and the final output layer—resulted in the best performance, as shown in Fig. 7. Additionally, the performance of the model was higher when the final layers of the model, which concatenate the outputs of both branches of the model, were frozen.

We found that base models constructed on the Guo eSpCas9 dataset had better transfer learning performance than base models constructed with the Guo SpCas9 dataset. TevSpCas9 average 5-fold cross validation performance improved by 0.053 rank correlation when transfer learning with the eSpCas9 base model versus the Guo SpCas9 base model (Supplementary Data S7). We found that crisprHAL with a dual branch architecture performed well on our TevSpCas9 dataset ($n = 279$) after transfer learning from a base model built on eSpCas9 data ($n = 45,010$), with a rank correlation of 0.630 (Fig. 7). This exceeds the best prior bacterial model, built for SpCas9 by Guo et al., which had a rank correlation in this dataset of 0.52 (Fig. 1A). Within the four prior bacterial models tested, we found that the gradient boosting

regression tree (GBR) models for SpCas9 and eSpCas9 by Guo et al. generalize to TevSpCas9 data better than the deep learning based models for SpCas9 and eSpCas9 from DeepSgRNA, respectively[29,35]. This contrasts with the improved performance from the deep learning models versus the GBR models on their own SpCas9 and eSpCas9 data; both Guo and DeepSgRNA construct their models on the same data.

To validate model performance, we tested crisprHAL on a set of unique sequences from the Guo et al. SpCas9 dataset ($n = 8728$)[29]. This set of sequences was curated to remove any overlap with the Guo eSpCas9 dataset used to construct the base model, a process which removed 36342 sgRNAs. As shown in Fig. 8B, crisprHAL performs well on this dataset with a rank correlation of 0.682.

### Applying transfer learning model to the SpCas9 dataset

We also tested the model with the SpCas9 dataset ($n = 303$) in place of TevSpCas9 for transfer learning from the eSpCas9 base model while leaving all other aspects of the model unaltered. The model performs well with the SpCas9 data, resulting in a 5-fold cross validation average rank correlation of 0.627 (Fig. 8C). This performance exceeds that of all existing models, with the best prior model, built for SpCas9 by Guo et al., attaining a rank correlation of 0.456 (Fig. 1B)[29,35]. We noted transfer learning to be an essential component of the SpCas9 performance. Without transfer learning the model reaches a rank correlation of only 0.417.

In line with our TevSpCas9 model performance, our SpCas9 model performs best when using the dual branch model, with performance only marginally exceeding that of the hybrid CNN-RNN alone (Fig. 7). Although performance is optimal when using the eSpCas9 dataset base model, we found models used our SpCas9 dataset to perform well when transfer learning from the Guo SpCas9 dataset base model, with a 5-fold cross validation average rank correlation of 0.609, notably higher than the TevSpCas9 results (Fig. 7). When testing SpCas9 model generalization, we found it to perform well on the unique Guo SpCas9 dataset, with a rank correlation of 0.657 (Fig. 8D)[29]. Testing could not be performed with the TevSpCas9 dataset due to the presence of all sgRNAs being cross-listed.

### crisprHAL predictions are generalizable to other bacteria

One significant issue with most existing Cas9/sgRNA prediction models is that they fail to generalize to datasets outside of those used for training[26]. We took two approaches to test if crisprHAL could accurately predict SpCas9/sgRNA activity on datasets generated for different bacteria. First, we designed a pool of 31,796 sgRNAs targeted against the chromosome of *Citrobacter rodentium* (Supplementary Data 9) that is used in mouse models of enterohaemorrhagic *E. coli* (EHEC) infections[56–58]. This experiment differed from the pTox experiment in that highly active Cas9/sgRNA combinations would become depleted relative to weakly active or inactive combinations because they would cleave the *C. rodentium* genome, promoting replication fork collapse and cell death (Supplementary Data 10). We found a rank correlation of 0.635 and 0.612 for TevSpCas9 measured activity versus the TevSpCas9 and SpCas9 crisprHAL model predicted activities (Fig. 8E, F). Second, we cloned a 2-kb fragment of the *katG* gene from *Salmonella enterica* Typhimurium LT2 into pTox to create pTox+KatG. We designed a pool containing 296 sgRNAs targeting the *katG* sequence (Supplementary Data 11), and measured their activity as described for the pTox/oPool experiment (Supplementary Data 12). We found rank correlations of 0.678 and 0.648 between the measured activity of the TevSpCas9/sgRNA and the TevSpCas9 and SpCas9 crisprHAL model predicted activities, respectively (Fig. 8G, H). Collectively, these experiments demonstrate that crisprHAL predictions can be generalized to bacteria other than *E. coli* and that predictions are robust to different measurements of SpCas9/sgRNA activity (depletion versus enrichment).

### Downstream target site nucleotides impact predictive performance

Prior models have proposed various input sequence lengths for optimal predictive performance[29,35]. For example, the DeepSgRNABacteria model suggests that 43nt input sequences may be optimal for eSpCas9 and SpCas9 performance based on calculated importance scores, with nucleotides downstream of the sgRNA binding site containing more information than upstream nucleotides[35]. Biologically, it is implausible that sequences outside of those contacted by either the sgRNA or the SpCas9 nuclease[59–63] should have a large effect on a machine learning model, and these models may thus be overparameterized.

To identify the optimal input sequence length to use for our model, we constructed versions of our model with input sequences extending upstream and downstream of the 20nt sgRNA target site (Fig. 9A). The 20-nt base input was extended upstream 10 positions in 1-nt increments upstream and downstream 11 positions in 1-nt increments. A single increment of 3 nt covered the PAM sequence. Predictive performance of these incremental models was measured by 5-fold CV across the TevSpCas9 dataset using rank correlation (Fig. 9B).

We noted that nucleotide additions upstream of the sgRNA target site immediately decreased the predictive ability of the model (Fig. 9B). In contrast, nucleotide additions downstream of the sgRNA target site improved predictive performance, up to the limit of 8nt downstream. Based upon these results we chose an input sequence of 28nt, comprising the 20 nt sgRNA target site, the 3nt PAM, and five additional downstream nucleotides. No upstream nucleotides were included in our input sequence for the crisprHAL model.

## Discussion

Although targeting SpCas9/sgRNA to desired sequences in small-sized bacterial genomes appears straightforward because it relies on apparent nucleotide complementarity, there are significant limitations in our ability to reliably identify highly active SpCas9/sgRNA combinations. Ideally, a predictive model of SpCas9/sgRNA activity should be agnostic to different datasets, generalize to different organismal systems, and recapitulate the known biology of SpCas9/sgRNA target interactions. Current prediction models do not meet all of these criteria. Here, we identify three areas that improve computational models of SpCas9/sgRNA activity; collection of biological data that accurately assesses SpCas9/sgRNA cleavage, appropriate treatment of high-throughput Illumina data for model training, and transfer learning to capitalize on existing and additional datasets.

Accurate computational predictions of sgRNA activity rely on biological data that reports on SpCas9/sgRNA cleavage activity and not secondary outcomes of DNA cleavage. This is particularly relevant in mammalian systems where many Cas9/sgRNA datasets report on non-homologous end joining (NHEJ) DNA repair outcomes of cleavage rather than directly assessing Cas9 cleavage. While bacteria generally lack NHEJ pathways, Cas9/sgRNA cleavage can be enhanced in *recA* deficient strains, or strains expressing dominant negative *recA* variants, to suppress DNA repair through the SOS response[30]. Our strategy to assess SpCas9/sgRNA cleavage was to use a two-plasmid enrichment assay with pooled libraries read out by Illumina sequencing that agrees well with the kinetics of in vitro DNA cleavage[48,64]. Crucially, this system also helps distinguish SpCas9/sgRNA on-target activity from toxicity because we measure both the relative activity and abundance of SpCas9/sgRNA combinations. Distinguishing toxicity from activity can be a confounding issue in bacterial systems where overexpression of Cas9 (or dCas9) can cause cellular toxicity, or at the very least to reduce the growth rate significantly. With the enrichment assay, we found that about one in seven SpCas9/sgRNA combinations showed evidence of toxicity. Our data suggest that there are different mechanisms of sgRNA-dependent toxicity, one of which results from sgRNAs with sufficient off-target identify to cleave the bacterial

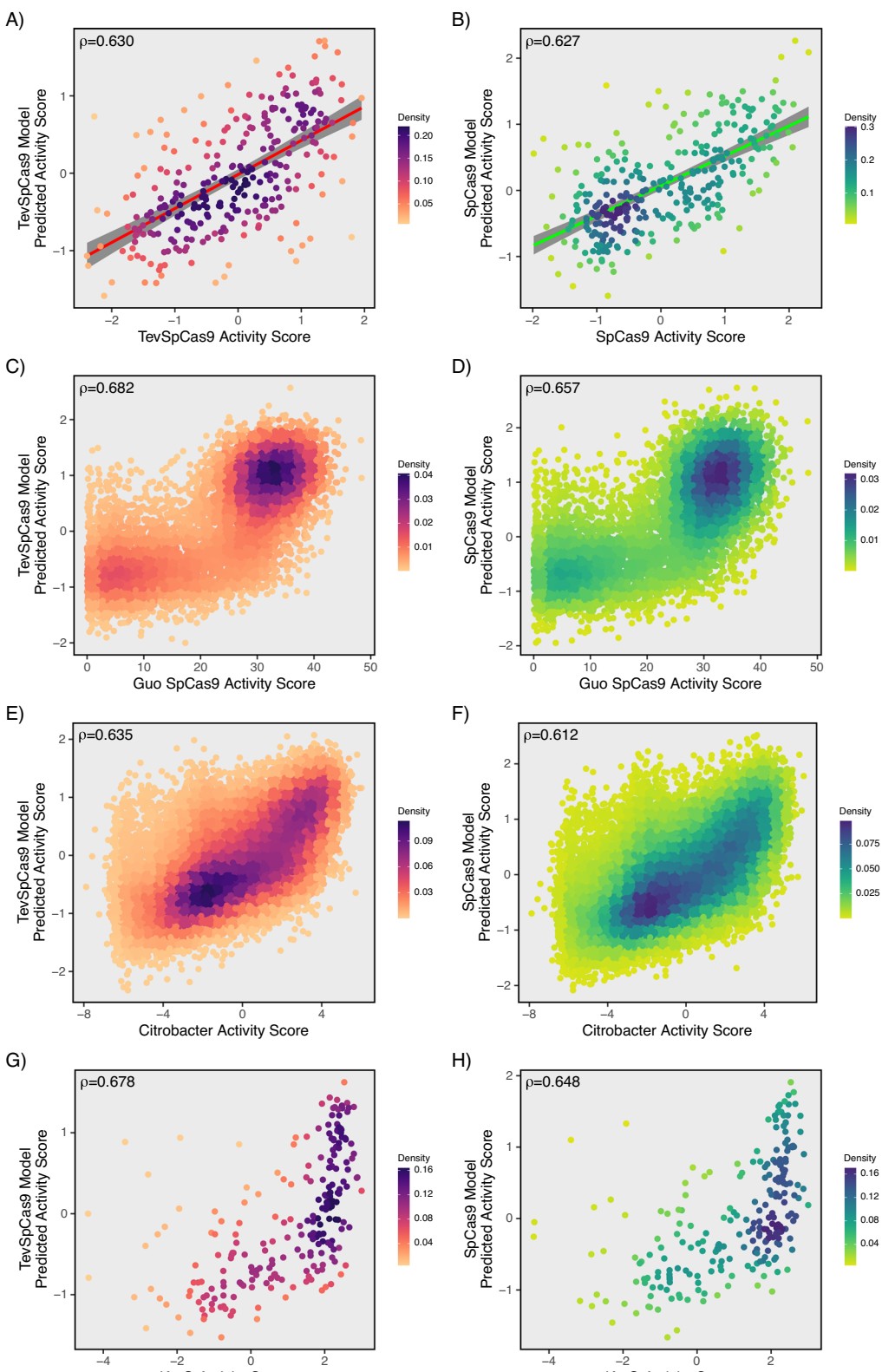

**Fig. 8 | Predictive performance of crisprHAL. A** Correlation between the TevSp-Cas9 dataset (*n* = 279) and the TevSpCas9 model 5-fold cross validation predictions (mean rank correlation of 0.630 across 5-folds). **B** Correlation between the SpCas9 dataset (*n* = 302) and the SpCas9 model 5-fold cross validation predictions (mean rank correlation of 0.627 across 5-folds). For (**A**) and (**B**) the line of best fit is indicated and the 95% confidence interval is represented by a gray shaded area.

**C** TevSpCas9 model and (**D**) SpCas9 model prediction correlations with the original Z-scores from the unique Guo SpCas9 dataset (*n* = 7821). **E** TevSpCas9 model and (**F**) SpCas9 model prediction correlations with the *Citrobacter rodentium* TevSp-Cas9 dataset (*n* = 30138). **G** TevSpCas9 model and (**H**) SpCas9 model prediction correlations with the *S. entericakatG* fragment TevSpCas9 dataset (*n* = 228). Source data are provided as a Source Data file.

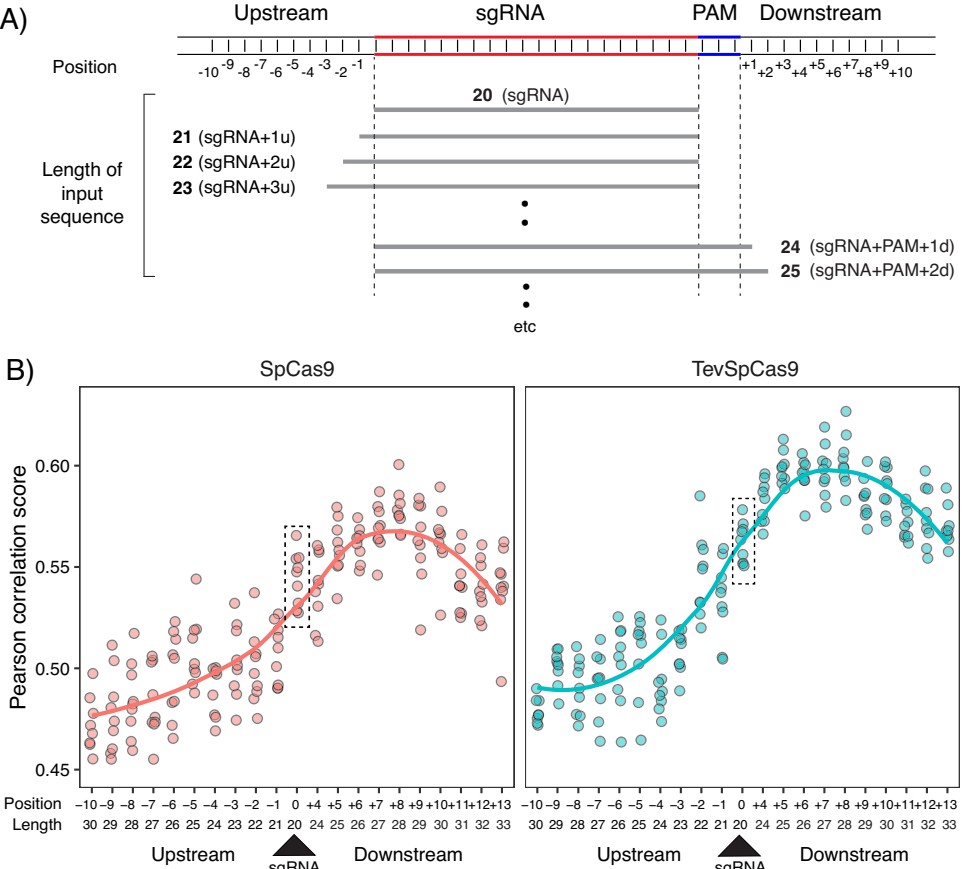

**Fig. 9 | Impact of sgRNA target site sequence length on model performance.**
**A** Length of sgRNA target site sequences tested as model inputs. All tested sequences include the 20nt sgRNA target site. Upstream (-) tests extend the model input sequence by 1nt 10 times. Downstream (+) tests begin with the addition of the NGG PAM to the model input, then extend by 1nt 10 times. **B** TevSpCas9 model

(left) and SpCas9 model (right) performance on their respective dataset versus the sgRNA sequence length input. Performance is measured by average rank correlation produced by 5-fold cross validation. Both models utilize transfer learning from our base model trained on the eSpCas9 dataset. The boxed regions in (**B**) represent the 20 nt sgRNA sequence. Source data are provided as a Source Data file.

chromosome. However, we also identified toxic sgRNAs that have multiple mismatches to chromosomal targets in positions that our mismatch profiling data indicate would not support SpCas9 cleavage. It is possible that toxicity could result from partial matches between the sgRNA and functionally critical genes on the chromosome that preclude DNA cleavage but facilitate transcriptional repression[65,66]. In large-scale pooled sgRNA depletion experiments, these sgRNAs would mistakenly be classified as having high on-target activity and add noise when training machine learning models. In more directed applications like the one implemented in this study, these sgRNAs are easily identifiable and no longer reported as false positives. Moreover, overtly toxic sgRNAs with high identity to chromosomal targets can be easily avoided during the design process[8].

One parameter for model inclusion that we explored in detail was the length of up- and down-stream sequence flanking the 20-nt sgRNA target site. Inclusion of flanking DNA sequence in prior models was justified by factors such as chromatin accessibility, consideration of DNA unwinding, and Cas9 activity data that indicated nucleotide preference in flanking regions (although it is possible this reflects DNA repair and not Cas9 cleavage preference). However, outside of the 20-nt sgRNA-target strand interaction, Cas9-target DNA contacts occur exclusively downstream of the PAM sequence, including a transient interaction 14-nt downstream that impacts binding and dissociation[59–63]. Thus, the biological data argue against inclusion of upstream DNA sequences. Indeed, our data show the best model performance with a 28-nt input sequence that includes the sgRNA

binding site, the PAM and 5 downstream nucleotides, and that inclusion of upstream sequence is uninformative.

Significantly, we found that the small amounts of high quality data generated in this study that while insufficient to train a model on their own, improved sgRNA activity predictions with the crisprHAL model that utilized machine transfer learning as well as a unique dual-branch CNN and RNN architecture. Our work corroborates prior findings that hybrid CNN-RNN architectures are well suited for transfer learning[43,67,68]. We found that the multi-layer CNN was the primary contributor to base model performance on the eSpCas9 data, reaffirming its use by models such as DeepSgRNA[35]. Inclusion of this multi-layer CNN branch, in addition to the hybrid CNN-BGRU, improved base model performance on eSpCas9 while retaining transfer learning capacity. Our dual branch structure provided a significant boost to model generalization performance on the unique Guo SpCas9 dataset as compared to the hybrid CNN-BGRU only architecture. Additionally, since all parameters in the multi-layer CNN and branch concatenation layers were frozen, nullification of the multi-layer CNN branch's contribution to the output prediction was unlikely. We attained the best model performance when using the same scoring method across the datasets, while compensating for variations in dynamic range through scaling by the standard deviation of scores from each dataset. With this treatment, crisprHAL predictions showed a linear correlation with measured sgRNA activity. Moreover, we did not utilize negative control sgRNAs in our process for sgRNA activity score calculations. Given that our eSpCas9 base model performs at least as well as the prior Guo

and DeepSgRNA models constructed on that dataset, we suggest that negative control sgRNAs are unnecessary for the scoring of SpCas9/sgRNA activity.

One issue with current Cas9/sgRNA prediction models is that they do not perform well outside of the initial datasets on which they were trained[26]. This lack of generalizability suggests that the training data is not accurately capturing Cas9/sgRNA cleavage activity, because there is no reason to believe that the biochemical basis of Cas9 cleavage should differ between organisms to impact predictive activity scores. Indeed, to our knowledge, there has been no attempt to test the generalizability of current sgRNA bacterial prediction models. Significantly, we showed that crisprHAL can predict with reasonable confidence activity for sgRNAs targeted to different bacteria (*S. enterica* and *C. rodentium*). Both of these bacteria have different codon usage and dinucleotide frequencies than E. coli, and the sgRNA sequences used are distinct from those in the training datasets. Thus, crisprHAL cannot be memorizing any aspect of the training datasets to make predictions for *S. enterica* or *C. rodentium*. Moreover, these experiments suggest that additional high-quality data for different species can be used for transfer learning with crisprHAL to improve predictions for a wide range of bacteria.

In summary, we have generated datasets for the activity of several hundred SpCas9/sgRNA and TevSpCas9/sgRNA combinations in a bacterial environment. The experimental setup detects activity over a large dynamic range and is able to distinguish toxicity from on-target cleavage activity. The datasets were then used in conjunction with machine transfer learning and a model architecture to produce crisprHAL, the most accurate TevSpCas9 and SpCas9 activity prediction model for bacteria to date. Our results show that small amounts of high-quality data can improve predictions of sgRNA activity and represent a step towards a generalizable model for bacteria. In principle, the approach outlined here to improve sgRNA activity predictions could be applied to any biological system where it is possible to collect high-quality Cas9/sgRNA cleavage data. When using in conjunction with existing large datasets for base model training, this will allow researchers to use transfer learning to fine-tune crisprHAL for their organism of choice by generating relatively small datasets that will overcome the barrier in research time and cost needed for both the deep sequencing experiments and the training of a model on those datasets. Overall, crisprHAL will enable more accurate prediction of sgRNAs for bacterial applications, including the use of Cas9/sgRNA as an antimicrobial agent, for eliminating plasmids or phages from strains, and for genome engineering of phage and bacterial genomes.

## Methods
### Bacterial strains
*E. coli* Epi300 (F'λ⁻mcrAΔ(mrr-hsdRMS-mcrBC)φ80d*lacZ* Δ *M15* Δ *(lac) X74 recA1 endA1 araD139* Δ *(ara,leu)7697 galU galK rpsL* (Str^R) *nupG trfA dhfr*) (Epicenter) was used for cloning the sgRNA pools. Screening sgRNA activity using a two-plasmid enrichment was done in NEB 5-alpha F'I^q *E. coli* (F' *proA*⁺*B*⁺*lacI^q*Δ*(lacZ) M15 zzf::Tn10(Tet^R) /fhuA2* Δ*(argF-lacZ)U169 phoA glnV44φ80*Δ *(lacZ)M15 gyrA96 recA1 relA1 endA1 thi-1 hsdR17*) strain harboring pTox. *Citrobacter rodentium* DBS100 was used for screening of sgRNA activity against chromosomal targets.

### Construction of sgRNA pools
pTox was screened for 5'-NGG-3' PAM sequences in a unique 3.2 kb region that included the kanamycin acetlytransferase (*kan^R*) coding region, the pBR322 origin of replication, and the *ccdB* DNA gyrase toxin coding region. The DNA sequence 20 nts upstream of each PAM site was computationally extracted to create a pool of 304 sgRNA with exact matches to pTox (oPool). We also included 15 sgRNAs with mismatches at various positions and 48 non-targeting sgRNAs as internal controls (Supplementary Data 3). To create the library of

sgRNAs with nucleotide transversions (mPool), 28 sgRNAs were picked from the oPool and single and double nucleotide transversions were tiled along the length of each oligo (Supplementary Data 1). Sequences that contain BsaI-HF-V2 restriction sites that generate correct overhangs for Golden Gate Cloning were added to the ends of the sgRNA sequences for subsequent cloning. The sequence 5'-CCTGGTTCTTGGTCTCTCACG-3' was added upstream of the sgRNA and 5'-GTTTTAGAGACCGCTGCCAGTTCATTTCTTAGGG-3' was added downstream to allow for efficient and directional cloning. Each pool was ordered as single-stranded fragments at 1 pmol/oligo from Integrated DNA technologies (IDT). For each library, second strand synthesis was performed using 1 μg of single stranded pool DNA and equimolar amounts of primer DE5224 in NEB buffer 2 (50 nM NaCl, 10 mM Tris-HCl, 10 mM MgCl₂, 1 mM DTT, pH 7.9) by denaturing at 94 °C for 5 min. Primers were annealed by decreasing temperature 0.1 °C/second to 56 °C and holding for 5 min, and followed by decreasing temperature 0.1 °C/second to 37 °C. To the annealed oligonucleotides, 1 μL of Klenow polymerase (New England Biolabs) and 1 μL of 10 mM dNTPs were added and incubated for 1 h at 37 °C, followed by a 20 min incubation at 75 °C before being held at 4 °C. The resulting dsDNA fragments were purified using a Zymogen DNA Clean & Concentrator-5 kit following manufacturer specifications. Golden Gate cloning was used to clone the oPool and mPool into SpCas9 and TevSpCas9 by combining 6 pmol of oPool or mPool, 100 ng of backbone plasmid, 0.002 mg BSA, 2 μL T4 DNA ligase buffer (50 mM Tris-HCl, 10 mM MgCl₂, 1 mM ATP, 10 mM DTT, pH 7.5), 160 units T4 DNA ligase (New England Biolabs) and 20 units of BsaI-HF-V2 (New England Biolabs) with the following thermocycler conditions: 37 °C for 5 min then 22 °C for 5 min for 10 cycles, 37 °C for 30 min, 80 °C for 20 min, 12 °C inf. The resulting pool was then transformed by heatshock into *E. coli* Epi300 and plated on LB plates (10 g/L tryptone, 5 g/L yeast extract, 10 g/L sodium chloride, 1% agar) supplemented with 25 mg/mL chloramphenicol and 0.2% w/v D-glucose.

To create pTox+KatG, a 2 kb fragment corresponding to the *Salmonella enterica* Typhimurium LT2 *katG* gene was amplified by PCR using primers DE6665 and DE6666 and cloned into pTox using Gibson Assembly (Supplementary Fig. S2). There were 296 sgRNA target sites in the *katG* fragment, 324 in the pTox backbone (303 of which are in the original pTox pool) and 20 non-targeting sgRNAs were included (Supplementary Data 11).

### Pooled sgRNA two-plasmid enrichment experiment
A two-plasmid enrichment experiment was used to assay sgRNA activity as previously described[45,46]. For liquid selections, 50 ng of the sgRNA plasmid pool was transformed into 50 μL E. coli NEB 5-alpha F'I^q competent cells harboring pTox by heat shock. Cells were allowed to recover in 1 mL of non-selective 2xYT media (16 g/L, 10 g/L yeast extract, and 5 g/L NaCl) for 30 min at 37 °C with shaking at 225 rpm. The recovery was then split and 500 μL was added to 500 μL of inducing 2xYT (0.04% (w/v) L-arabinose and 50 mg/mL chloramphenicol) or to 500 μL of repressive 2xYT (0.4% (w/v) D-glucose and 50 mg/mL chloramphenicol) and incubated for 90 min at 37 °C with shaking at 225 rpm. The two cultures were washed with 1 mL of inducing media (1x M9, 0.8% (w/v) tryptone, 1% v/v glycerol, 1 mM MgSO₄, 1 mM CaCl₂, 0.2% (w/v) thiamine, 10 mg/mL tetracycline, 25 mg/mL chloramphenicol, 0.4 mM IPTG) or repressed media (1x M9, 0.8% (w/v) tryptone, 1% v/v glycerol, 1 mM MgSO₄, 1 mM CaCl₂, 0.2% (w/v) thiamine, 10 mg/mL tetracycline, 25 mg/mL chloramphenicol, 0.2% (w/v) D-glucose) respectively before addition to 50 mL of the same media that was used in the wash in a 250 mL baffled flask. These cells were grown overnight at 37 °C with shaking at 225 rpm. Plasmids were then isolated using the Monarch Plasmid Miniprep Kit (NEB) according to manufacturers specifications. The sgRNA locus was then PCR amplified using primers (Supplementary Table S3) containing Ilumina adapter sequence, four random nucleotides, 12-mer barcodes to specify the

replicate, and plasmid-specific nucleotides at the 3′ end. The resulting amplicons were sent for 150 bp paired-end Illumina MiSeq sequencing at the London Regional Genomics Center (London, ON).

## Growth-curve experiments with individual sgRNAs

The pool of cells containing pTevSpCas9+sgRNA was grown overnight in selective LB (25 mg/mL chloramphenicol and 0.2% (w/v) D-glucose), diluted and plated on agar plates (10 g/L tryptone, 5 g/L yeast extract, 10 g/L sodium chloride, 1.5% agar (w/v) supplemented with 25/mL chloramphenicol and 0.2% w/v D-glucose). Individual colonies were selected and grown overnight in selective LB (25 mg/mL chloramphenicol and 0.2% (w/v) D-glucose) before plasmids were isolated using the Monarch Plasmid Miniprep Kit (NEB) according to manufacteur specificatons. The sgRNA locus of each plasmid was Sanger sequenced at London Regional Genomics Center (London, ON) to determine the sgRNA identity. In three independent transformations, 20 ng of each plasmid, isolated oPool DNA, and pTevSpCas9 with no sgRNA were transformed into 20 μL *E. coli* competent NEB 5-alpha F′I$^q$ cells harboring the pTox. Cells were allowed to recover in 1 mL of non-selective 2xYT media (16 g/L, 10 g/L yeast extract, and 5 g/L NaCl) for 30 min at 37 °C with shaking at 225 rpm. The recovery was then split and 500 μL was added to 500 μL of inducing 2xYT (0.04% (w/v) L-arabinose and 50 mg/mL chloramphenicol) or to 500 μL of repressive 2xYT (0.4% (w/v) D-glucose and 50 mg/mL chloramphenicol) and incubated for 40 min at 37 °C with shaking at 225 rpm. These cultures were then plated on inducing or repressing M9 plates and grown overnight at 37 °C. At the same time, 20 μL was added to 180 μL of inducing and repressing M9 liquid media in a 96-well plate for growth curves. Plates were grown at 37 °C in the BioTek Epoch 2 Microplate Spectrophotometer measuring the absorbance at 600 nm every 10 min for 18 h with double orbital shaking. Raw data was collected, processed, and analyzed using the Growthcurver R package[69].

## *Citrobacter rodentium* sgRNA pool construction and depletion assay

A 236-kb fragment of the *Citrobacter rodentium* DBS100 genome was screened for 5′-NGG-3′ PAM sequences and 31,596 sites were identified (Supplemental Dataset 6). The 20 bp upstream of each PAM was extracted and ordered as a pool from Twist Bioscience after appending the sequence 5′-CCTGGTTCTTGGTCTCTCACG-3′ upstream of the sgRNA and 5′-GTTTTAGAGACCGCTGCCAGTTCATTTCTTAGGG-3′ downstream for cloning. The pool also contained 200 non-targeting sgRNAs(Supplemental Dataset 6). The sgRNA pool was made double stranded by PCR amplification with primers DE5231 and DE5224 (Supplementary Table S3) using 1 ng of single stranded sgRNA template and cloned into pTevSpCas9 plasmid as described above. Five independent cloning reactions were electroportated into *E. coli* Epi300 in and added to 500 mL of LB (10 g/L tryptone, 5 g/L yeast extract, 10 g/L sodium chloride) supplemented with 25 mg/mL chloramphenicol and 0.2% w/v D-glucose to grow at 37 °C with shaking at 225 rpm overnight. The plasmid pool was then isolated using the Monarch Plasmid Miniprep Kit (NEB) according to manufacturer's specifications.

In 10 independent reactions, 50 ng of the sgRNA pool was electroporated into 100 μL *C. rodentium* competent cells and allowed to recover in 1 mL of non-selective 2xYT media (16 g/L, 10 g/L yeast extract, and 5 g/L NaCl) for 30 min at 37 °C with shaking at 225 rpm. The recovery was then split and 500 μL was added to 500 μL of inducing 2xYT (0.4%) (w/v) L-arabinose and 50 mg/mL chloramphenicol) or to 500 μL orepressive 2xYT (0.4%) (w/v) D-glucose and 50 mg/mL chloramphenicol) and incubated for 90 min at 37 °C with shaking at 225 rpm. The resulting cultures were then added to 50 mL of LB (10 g/L tryptone, 5 g/L yeast extract, 10 g/L sodium chloride) supplemented with 25 mg/mL chloramphenicol and 0.2% w/v D-glucose and grown overnight at 37 °C with shaking at 225 rpm. Plasmids were

isolated using the Monarch Plasmid Miniprep Kit (NEB) and the sgRNA locus was then PCR amplified using primers (Supplementary Table S3) containing Ilumina adapter sequence, four random nucleotides, 12-mer barcodes, and plasmid-specific nucleotides at the 3′ end. The resulting amplicons were sent for 150 bp paired-end Illumina NextSeq High Output sequencing the London Regional Genomics Center (London, ON).

## Datasets and input sequence encoding

Two distinct groups of data are utilized in model development: data generated in this study using the nuclease TevSpCas9 and sgRNAs targeted to the pTox plasmid, and a published dataset using the eSp-Cas9 and SpCas9 nucleases with ~70,000 sgRNAs targeted to the *E. coli* genome from Guo and colleagues[29]. Due to the methodology in the Guo study, the eSpCas9 and SpCas9 datasets contain overlapping sgRNA target sequences. To generate a unique sgRNA testing set for model testing, sgRNAs in the Guo SpCas9 dataset that are cross-listed with the eSpCas9 dataset were removed. We refer to this dataset as the unique sgRNA Guo SpCas9 dataset. All sgRNAs were mapped to the *E. coli* genome and pTox plasmid and sgRNAs with ≥15nt PAM proximal matches to an off-target site were excluded from our datasets. Based on these mapping results, 43nt target site sequences were obtained for each sgRNA and containing the 20nt sgRNA target site, the 3nt PAM, and 10nt upstream and downstream of the sgRNA target site. These extended inputs provided the ability to test sequence length versus predictive performance.

The nucleotides comprising the sgRNA target site sequences are commonly represented with strings of single characters (A, C, G, T) each representing a nucleotide. However, alphabetic encoding of nucleotides is not useful for deep learning models. We converted our input sequences with one-hot encoding, where the input sequence is represented as a 4-by-N matrix – 4 nucleotide options across an N-length input sequence. The nucleotides, A, C, G, and T, are encoded as [1 0 0 1], [0 1 0 0], [0 0 1 0], and [0 0 0 1] respectively.

## Data processing and activity score calculation

Reads from the Illumina sequencing were parsed using a custom script that deconvolute the barcoded sequences into a table that contained replicates of induced or repressed conditions. The bacterial sgRNA read counts from these datasets representing on-target activity scores are compositional in nature[55], and therefore require normalization or transformation to become interpretable[70]. All sgRNAs in the Guo et al. datasets having a read count less than 20 in either replicate of the catalytically dead Cas9 (eSpdCas9 and SpdCas9) samples were removed. Relative abundance ('rab.all') and difference values ('diff.btw') for each guide were calculated using the 'aldex.effect' function of ALDEx2[55]. For the *C. rodentium* depletion datatset, we used the initial read count from the sgRNA sequencing pool prior to transformation in place of replicates for the repressed condition as the input to ALDEx2. For the *S. entericakatG* dataset, highly variable guides with a difference within (ALDEx2 'diff.win') >1 were removed from testing. Scores used in model training were then normalized to generate the final activity scores by dividing each value by the standard deviation from its respective dataset (Supplementary Fig. S8). To obtain an untouched dataset for model generalization testing, we used the original, Z-score based normalization, sgRNA activity scoring by Guo et al. for the unique Guo SpCas9 dataset[29]. Data were plotted using R.

## Model construction and transfer learning

During model development we tested various architectures, including those with multiple branches, to test the performance of CNN and RNN neural networks. A CNN is an artificial neural network which excels at capturing spatial information from an input. This capability results in the frequent application of CNNs to image recognition problems.

Similar to pixels in an image, one-hot encoded nucleotide sequences can be used as inputs to a CNN, whereby local nucleotide preferences can be extracted[35,43,68].

Contrasting the CNNs local information capture capabilities, RNNs excel at learning sequential information. RNNs contain an internal memory state which are updated to learn important interactions within a sequence. Prior work has shown the benefit of utilizing a combination of CNN and RNN layers within a model to improve performance[43,68]. Spatial information captured by CNN layers can be fed to the RNN, whereby sequential information is then deduced, increasing performance[67].

We developed models on prior datasets to optimize for transfer learning – referred to as base models. Transfer learning is a method whereby a model utilizes information transferred from a similar domain to improve performance[71]. In practice, the base model is commonly constructed on datasets larger than those to which the transfer learning will be applied. For our context, we test models constructed on either the Guo SpCas9 or eSpCas9 dataset, and apply those base models as the starting point for training on our smaller datasets. To maximize the benefit of the pre-learned information from the base model, we tested variations in model layer freezing, where parameters in specific layers of the model are fixed before transfer learning model training occurs.

### Model training and tuning
We constructed crisprHAL with Tensorflow Keras[72]. This network was trained using the optimizer Adam, with mean squared error used as the loss function. The transfer learning model was tuned using 5-fold cross validation with a 80% training set and a 20% test set for each fold. The base model was tuned using a simple 80% training and 20% testing set split. Hyperparameter tuning was performed for a number of factors affecting the model, including: number of CNN layers, number of dense layers, channel sizes, CNN window sizes, RNN size, dense layer sizes, dropout rates, and activation functions between layers. Base model epochs were optimized by testing in increments of 5, versus transfer learning epochs, which were tested in single epoch increments. During hyperparameter tuning, the following activation functions were tested: linear, sigmoid, tanh, ReLU, and LeakyReLU. Batch sizes were optimized separately for the base and transfer learning model stages. Smaller batch sizes for transfer learning model training were preferential due to the smaller datasets used and the greater importance of accuracy in this stage, relative to the base model. Architecture testing of variations in the total number of CNN and/or RNN layers within the model used homogeneous hyperparameters for each type of neural network layer.

### Installation and testing of other models
We installed and ran the Guo and DeepSgRNA models (downloaded from Github sites https://github.com/zhangchonglab/sgRNA-cleavage-activity-prediction and https://github.com/biomedBit/DeepSgrnaBacteria)[29,35]. To test the Guo SpCas9 and eSpCas9 models, we converted our sgRNA-associated target site sequence inputs to the required 30nt length, containing the 20nt sgRNA target, 4nt upstream, and 6nt downstream including the NGG PAM. To test the DeepSgRNA SpCas9 and eSpCas9 models, we converted our sgRNA-associated target site sequence inputs to the required 43nt length, containing the 20nt sgRNA target, 10nt upstream, and 13nt downstream including the NGG PAM. The eukaryotic sgRNA prediction models DeepHF[40], C-RNNCrispr[41], DeepSpCas9[42] and Crispr-NET[43] were downloaded, installed and used to predict the activities of the oPool sgRNAs targeted to pTox. We compared the predicted activities to the measured activities for TevSpCas9 and SpCas9 using Spearman rank correlation. We also downloaded and installed DeepGuide[44], retrained the model on the Guo eSpCas9 dataset, and tested predicted versus measured activity for the pTox TevSpCas9 dataset.

### Performance and evaluation of models
To evaluate our models we used Spearman rank correlation coefficient, referred to as rank correlation. We chose this metric rather than Pearson correlation coefficient as it does not depend on a linear association between variables. Additionally, given its past use, it provides a clear metric from which to compare our models' performance to prior models[29,35,43,68]. We calculated rank correlation with the "spearmanr" function from the Scipy stats Python package[73].

### Statistics and reproducibility
We chose a sample-size based largely on convenience. The number of samples per group for the positive and negative selection experiments was large enough to identify the majority of significant features following the guidance of ref. 74 Significance was determined using the ALDEx2 R package[55,75] with an expected FDR of 0.05. All difference and dispersion measures were Expected values calculated from Bayesian posterior estimates of the sequencing data. This has been found to be more reproducible than using point estimate measures[55].

### Reporting summary
Further information on research design is available in the Nature Portfolio Reporting Summary linked to this article.

## Data availability
The Illumina sequencing datasets generated in this study have been deposited in the Sequence Read Archive with the accession code PRJNA939699. Source data is available as a Source Data file. Source data are provided with this paper.

## Code availability
Our model to predict TevSpCas9 and SpCas9 target site activity is available for download at https://github.com/tbrowne5/crisprHAL without restriction.

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

## Acknowledgements
Supported by a CIHR Project Grant (PJT-159708) to D.R.E. and G.B.G., D.T.H. and T.S.B. were supported by a MITACS Accelerate Award. We thank Michael Hallet for discussions on machine learning.

## Author contributions
D.T.H, T.S.B, G.B.G. and D.R.E. conceived the experiments, D.T.H, P.N.B, T.S.B conducted the experiments, D.T.H, T.S.B, G.B.G. and D.R.E. analyzed the results. All authors reviewed the paper.

## Competing interests
The authors declare no competing interests.
