## [Peer Review File · Nature Communications]

Reviewers' Comments:

Reviewer #1:

Remarks to the Author:

In this paper, the authors designed a two-plasmid positive selection system that can characterize the Cas9 cleavage activity. In addition, the authors proposed a deep-learning architecture (crisprHAL) that can predict the experiment results through transfer learning.

Major:

1. The experiment contained confounding variables. For both mPool and oPool profiling, it's better to use SpdCas9 and TevSpdCas9 to measure the effects of binding on cell growth/enrichment/survival rate. Especially for the oPool (Fig. 4A), as some guides were designed to target the ccdB coding region.

2. The authors' design cannot ensure to exclude off-target effects (no ≥ 15 nt PAM proximal matches to targets in genome), as it's commonly accepted that the PAM-proximal 7-12 nt region is regarded as the seed region (Cong et al., 2013, Boyle et al., 2017). Also, according to the author's results (Fig. 3B), some guides harboring single mismatches still showed significant activities. We here list some potential off-target hits below:

1) Non-template strand:

Sequence (oPool)	Off-target hits (genome)	Gene
GGGAGAGGCGGTTTGCGTAT	GGGAGAGGCGGTTTGCGTAT	lacI
GGCCAACGCGCGGGGAGAGG	GGCCAACGCGCGGGGAGAGG	lacI
ATCGGCCAACGCGCGGGGAG	ATCGGCCAACGCGCGGGGAG	lacI
TTAATGAATCGGCCAACGCG	TTAATGAATCGGCCAACGCG	lacI
GTCAACCTGCATTAATGAAT	TGCCAGCTGCATTAATGAAT	lacI
ATAACGAATGCGCCCGACGC	TTCCCCGGTGCGCCCGACGC	astC
CTACGGGGTCTGACGCTCAG	GAAGGTAGTCTGACGCTCAG	deoD
GATTAAGTTGGGTAACGCCA	CATGGCGCTGGGTAACGCCA	lacZ
CGATTAAGTTGGGTAACGCC	CAGCATGATTGGGTAACGCC	lacZ
TGCTGCAAGGCGATTAAGTT	AGATCAGTGGCGATTAAGTT	lacZ
CGAAAGGGGATGTGCTGCA	TCTTTAATGGATGTGCTGCA	lacZ
TTACGCCAGCTGGCGAAAGG	GCGATGATGCTGGCGAAAGG	lacZ
ATTACGCCAGCTGGCGAAAG	GAACAAAGAGCTGGCGAAAG	lacZ
TATTACGCCAGCTGGCGAAA	CACAGGCTCAGCTGGCGAAA	lacZ
CTATTACGCCAGCTGGCGAA	GTCCGTACCCAGCTGGCGAA	lacZ
CTCTTCGCTATTACGCCAGC	AGGGTTATTATTACGCCAGC	lacZ
TTGGGAAGGGCGATCGGTGC	TTGGGAAGGGCGATCGGTGC	lacZ
GTTGGGAAGGGCGATCGGTG	GTTGGGAAGGGCGATCGGTG	lacZ
CAACTGTTGGGAAGGGCGAT	CAACTGTTGGGAAGGGCGAT	lacZ
GGCTGCGCAACTGTTGGGAA	GGCTGCGCAACTGTTGGGAA	lacZ
AGGCTGCGCAACTGTTGGGA	AGGCTGCGCAACTGTTGGGA	lacZ
GTATAGGCTGCGCAACTGTT	ATTCAGGCTGCGCAACTGTT	lacZ
CGTATAGGCTGCGCAACTGT	CATTCAGGCTGCGCAACTGT	lacZ
GGGATCACCATCCGTCGCCC	GATCTCTTCATCCGTCGCCC	aer
ACATGGATGCTGATTTATAT	AACTTTCTGCTGATTTATAT	ygcB
CCCCGGGAAAACAGCATTCC	CACAAAGGAAAACAGCATTCC	wcaM
ACCGGATAAGGCGCAGCGGT	CCAGTTTCAGGCGCAGCGGT	aceK
GCAGCGGTGCGGCTGAACGG	GCAAAAAACGGGCTGAACGG	hyfR

2) template strand:

Sequence (oPool)	Off-target hits (genome)	Gene
GTTAAAAAAGTTGACGTAAC	AGGGGGCCAGTTGACGTAAC	def
ACGCGGCTTGGCGAACCGGA	ATTCATCGTGGCGAACCGGA	eutK
ACGCGGCTTGGCGAACCGGA	ATTCATCGTGGCGAACCGGA	eutK
GAATCGAATGCAACCGGCGC	CTGGTCGCTGCAACCGGCGC	nuoG

AAGACACGACTTATCGCCAC CAGAGCGAACTTATCGCCAC yqhD
CCCGACCGCTGCGCCTTATC GGATGCGGCTGCGCCTTATC ycjX
AGCATAAAGTGAAAGCCTG AGCATAAAGTGAAAGCCTG rlpA
AAGCATAAAGTGAAAGCCT AAGCATAAAGTGAAAGCCT rlpA
GAAGCATAAAGTGAAAGCC ACCCTTATAAGTGAAAGCC glnD
ATTCCACACAACATACGAGC ATTCCACACAACATACGAGC rlpA
CTATTACGCCAGCTGGCGAA GTCCGTACCCAGCTGGCGAA acrB
CTCTTCGCTATTACGCCAGC AGGGTTATTATTACGCCAGC norV
CTTTCGTTTTATTTGATGCC CTTTCGTTTTATTTGATGCC yccS
CTTTCGTTTTATTTGATGCC CTTTCGTTTTATTTGATGCC yccS
CAAACAACAGATAAAAACGAA CCAGCACCCAGATAAAAACGAA ydcO
CAAACAACAGATAAAAACGAA CCAGCACCCAGATAAAAACGAA ydcO
CGCAGCGGTGCGGGCTGAACG GCTTATTGTCGGGCTGAACG yagG
TATCCGGTAAGCGGCAGGGT GGTGGGCGAAGCGGCAGGGT hslR
TGTGAGTTAGCTCACTCATT TGTGAGTTAGCTCACTCATT yicO
GCTCACTCATTAGGCACCCC GCTCACTCATTAGGCACCCC yicO
GGCTTTACTTTTATGCTTC GTTAATCGACTTTATGCTTC yebQ
CTTCCGGCTCGTATGTTGTG CTTCCGGCTCGTATGTTGTG yicO
AGCGAATAACAATTTACACAGCGGATAACAATTTACACAG yicO
CGTCGTGACTGGGAAAACCC GAAGATAACTGGGAAAACCC hyfH
CACATCCCCCTTTCGCCAGC CCTTCACCCCTTTCGCCAGC ybaT
CAGGCATCAAATAAAAACGAA GAAAAAAGAAATAAAAACGAA efeB

3. Is that possible that the differences in growth curves were due to the leaky gene expression?

4. When froze the CNN layers, the trainable variables dramatically decreased. Therefore, is that possible that the increased performances were due to the avoidance of the overfitting problem instead of transfer learning? In addition, have the authors tried using resnet blocks in the CNN layers?

Minor:

1. In Fig. 1A, the text "growth curve" was obscured by the figure.

Reviewer #2:

Remarks to the Author:

The paper by Edgell and colleagues deals with the problem of predicting on-target activity of Cas9 sgRNA guides. The authors claim that "current bacterial sgRNA activity models struggle with accurate predictions and do not generalize well", which is true. The lack of generalization in those models is because they are species-specific (mostly designed for humans or other model organisms) and there is not enough data for *E. coli*. Quoting the authors: "the evidence indicates that there is a pressing need for additional high-quality bacterial sgRNA activity data sets to validate and generalize previous findings, and to provide training data for predictive machine learning models". Based on my experience, I agree with that statement. They also point out that "a complicating factor in assessing sgRNA activity in bacteria is that expression of Cas9 (and dCas9) alone can result in cellular toxicity and slow growth", which complicates the determination of what is a good cutter vs. a bad cutter.

The authors propose a method to generate high-quality training data using a two-plasmid positive selection system. The sgRNA activity can be analyzed by deep sequencing of the sgRNA expression cassette.

First, they test the Guo model [ref 28] and the DeepSgRNA model [ref 34] on their data. Their predictive performance of these models was modest (Spearman ~ 0.5). After explaining and analyzing comprehensively their new two-plasmid survival assay to measure sgRNA activity, they introduce a new deep learning model (crisprHAL) that predict sgRNA Cas9 activity in *E. coli*. crisprHAL was trained on Guo's SpCas9 or Guo's eSpCas9 datasets [ref 28] and when tested on

their data, Spearman increased from 0.52 to 0.62

Strengths:

- * The two-plasmid survival assay is clever, and it can distinguish highly active sgRNA guides from toxic ones with poor growth, even in repressed conditions
- * The assay was validated extensively
- * The assay's sensitivity was analyzed via a multiplexed sequencing experiment by introducing single and double nucleotide substitutions: sgRNA with no mismatches had higher activity than sgRNAs with one mismatch which in turn had higher activity than sgRNA with two mismatches
- * They showed that growth curves of individual sgRNAs can identify toxic guides: several sgRNAs that would be considered active based on their relative abundance in sequencing experiments had indeed high levels of toxicity
- * Overall the proposed methodology is sound
- * crisprHAL achieves rank correlation of 0.630 when trained on the Guo's dataset and tested on their dataset, better than the Guo model prediction which had a rank correlation of 0.52 (Figure 1)
- * The paper is well written, with high quality figures; there is enough details to reproduce the results

Weaknesses:

- * The dataset produced with this new assay is very very small; 279 guides for TevSpCas9 and 302 for SpCas9. This is completely inadequate to train a deep learning model. The community needs LARGE high-quality sgRNA activity datasets. Can this assay scale to produce much bigger datasets?
- * How many parameters does crisprHAL has? Figure 7A shows a complex architecture with many many layers. Considering that Guo's training set has 40-45 thousand guides (which is not very many for deep learning), how do you know that your model is not over-fitting and just "storing" the guides encoded as weights in the network? if so, the model would not be able to generalize at all -- which is one of your stated objectives
- * It would be interesting to compare pros and cons of the two-plasmid survival assay to the functional screens developed for non conventional yeasts in doi:10.1038/s41467-022-28540-0 and doi:10.1016/j.ymben.2019.06.007;
- * It would be informative to compare the predictive ability of DeepGuide (doi:10.1038/s41467-022-28540-0) with crisprHAL, but training DeepGuide on the Guo's dataset and testing it on your new dataset
- * Supplementary Tables S1-S4, which were given to me as hundreds of pages in PDF format, are unreadable and would not be useful to anyone in this format. I strongly suggest distributing them as Excel files.

Reviewer #3:

Remarks to the Author:

In this manuscript, the authors claim that they have developed an sgRNA bacterial activity prediction tool. This work is solid with interesting results. However, it looks to me that the progress is incremental for this high-impact journal, and its consideration for publication in Nature Communications would be premature as it. The following comments are made to strengthen the manuscript.

1. Recently, there have been many papers that report the use of an sgRNA bacterial activity prediction tool or its development. Although they discuss the report by Guo et al., I find that multiple recent papers are missing. For the balanced introduction and discussion sections, I highly recommend that the authors do an extensive literature search and cite relevant papers that refine the existing models or use such tools.
2. To make this work novel and impactful, the generalizability should be better tested and demonstrated. To this end, I highly suggest that they perform new experiments to include more data from different representative species and genera. Without such results, their work looks incremental.
3. Toxicity is a very complex phenotype, and the CRISPR toxicity has yet to be elucidated. It is known that there are big differences in Cas/gRNA toxicity between different species or genera. Thus, along with the above comment, the authors should provide more data by testing their systems in more species and genera.
4. Many different types of sgRNA prediction tools have been developed, including off-target effect prediction, strain-level specificity prediction, and activity prediction. Given that tools are eventually used for practical applications, I highly recommend that the authors discuss the usage of their 'generalizable' tools from the application perspective.

Response to reviewer comments for

A generalizable Cas9/sgRNA prediction model using machine transfer learning with small high-quality datasets (NCOMMS-23-12560-T)

Dalton T. Ham^{1,+}, Tyler S. Browne^{1,+}, Pooja N. Banglorewala¹, Tyler L. Wilson², Richard K. Michael², Gregory B. Gloor^{1,*}, David R. Edgell^{1,*}

¹Department of Biochemistry, Schulich School of Medicine and Dentistry, London, ON, N6A5C1, Canada

²Tesseraqt Optimization Inc, Toronto, ON, Canada

+These authors contributed equally to this work

*Corresponding authors: dedgell@uwo.ca, ggloor@uwo.ca

Reviewer Comments

Reviewer #1 (Remarks to the Author):

In this paper, the authors designed a two-plasmid positive selection system that can characterize the Cas9 cleavage activity. In addition, the authors proposed a deep-learning architecture (crisprHAL) that can predict the experiment results through transfer learning.

Major:

1. The experiment contained confounding variables. For both mPool and oPool profiling, it's better to use SpdCas9 and TevSpdCas9 to measure the effects of binding on cell growth/enrichment/survival rate. Especially for the oPool (Fig. 4A), as some guides were designed to target the *ccdB* coding region.

RESPONSE: We thank the reviewer for this comment but respectfully disagree that this is an appropriate control. Cas9 has different reaction kinetics than the inactive dCas9 (or TevdCas9) enzymes. While the association rate for Cas9 and dCas9 are similar, the dissociation rate for dCas9 is much longer than for Cas9. Studies in *E. coli* (Jones et al. Kinetics of dCas9 target search in *Escherichia coli*. Science 2017:1420-1424) suggest that dCas9 remains bound at its target site until the next round of DNA replication, whereas active Cas9 would dissociate after cleavage. Thus, we would anticipate confounding dCas9 effects on cell growth independent of cleavage for the same set of sgRNAs used in an active Cas9 experiment. Moreover, many of the sgRNAs are not targeted to the coding strand of *ccdB* within the promoter or first 200 nucleotides of the AUG codon, which we and others have determined to have maximal gene repression. Rather, we maintain that a paired experimental control where SpCas9 (or TevSpCas9) is either repressed by glucose or expressed with arabinose most accurately reports on differences in activity.

This is not to say that dCas9 data are not useful, they most certainly are since the base model of crisprHAL is trained on such, rather that the combination through transfer learning of datasets of different provenance results in a more accurate and generalizable model. We believe that the demonstration of our model in an orthogonal organism and system is a strong validation of this conclusion.

2. The authors' design cannot ensure to exclude off-target effects (no ≥ 15 nt PAM proximal matches to targets in genome), as it's commonly accepted that the PAM-proximal 7-12 nt region is regarded as the seed region (Cong et al., 2013, Boyle et al., 2017). Also, according to the author's results (Fig. 3B), some guides harboring single mismatches still showed significant activities. We here list some potential off-target hits below:

1) Non-template strand:

Sequence (oPool)	Off-target hits (genome)	Gene
GGGAGAGGCGGTTTGCGTAT	GGGAGAGGCGGTTTGCGTAT	lacI
GGCCAACGCGCGGGGAGAGG	GGCCAACGCGCGGGGAGAGG	lacI
ATCGGCCAACGCGCGGGGAG	ATCGGCCAACGCGCGGGGAG	lacI
TTAATGAATCGGCCAACGCG	TTAATGAATCGGCCAACGCG	lacI
GTCAACCTGCATTAATGAAT	TGCCAGCTGCATTAATGAAT	lacI
ATAACGAATGCGCCCGACGC	TTCCCCGGTGCGCCCGACGC	astC
CTACGGGGTCTGACGCTCAG	GAAGGTAGTCTGACGCTCAG	deoD
GATTAAGTTGGGTAACGCCA	CATGGCGCTGGGTAACGCCA	lacZ
CGATTAAGTTGGGTAACGCC	CAGCATGATTGGGTAACGCC	lacZ
TGCTGCAAGGCGATTAAGTT	AGATCAGTGGCGATTAAGTT	lacZ
CGAAAGGGGGATGTGCTGCA	TCTTTAATGGATGTGCTGCA	lacZ
TTACGCCAGCTGGCGAAAGG	GCGATGATGCTGGCGAAAGG	lacZ
ATTACGCCAGCTGGCGAAAG	GAACAAAGAGCTGGCGAAAG	lacZ
TATTACGCCAGCTGGCGAAA	CACAGGCTCAGCTGGCGAAA	lacZ
CTATTACGCCAGCTGGCGAA	GTCCGTACCCAGCTGGCGAA	lacZ
CTCTTCGCTATTACGCCAGC	AGGGTTATTATTACGCCAGC	lacZ
TTGGGAAGGGCGATCGGTGC	TTGGGAAGGGCGATCGGTGC	lacZ
GTTGGGAAGGGCGATCGGTG	GTTGGGAAGGGCGATCGGTG	lacZ
CAACTGTTGGGAAGGGCGAT	CAACTGTTGGGAAGGGCGAT	lacZ
GGCTGCGCAACTGTTGGGAA	GGCTGCGCAACTGTTGGGAA	lacZ
AGGCTGCGCAACTGTTGGGA	AGGCTGCGCAACTGTTGGGA	lacZ
GTATAGGCTGCGCAACTGTT	ATTCAGGCTGCGCAACTGTT	lacZ
CGTATAGGCTGCGCAACTGT	CATTCAGGCTGCGCAACTGT	lacZ
GGGATCACCATCCGTCGCC	GATCTCTTCATCCGTCGCC	aer
ACATGGATGCTGATTTATAT	AACTTTCTGCTGATTTATAT	ygcB
CCCCGGGAAAACAGCATTCC	CACAAAGGAAACAGCATTCC	wcaM
ACCGGATAAGGCGCAGCGGT	CCAGTTTCAGGCGCAGCGGT	aceK
GCAGCGGTCGGGCTGAACGG	GCAAAAACGGGCTGAACGG	hyfR

2) template strand:

Sequence (oPool) Off-target hits (genome) Gene

GTTAAAAAAGTTGACGTAAC AGGGGCCAGTTGACGTAAC def
ACGCGGCTTGGCGAACCGGA ATTCATCGTGGCGAACCGGA eutK
ACGCGGCTTGGCGAACCGGA ATTCATCGTGGCGAACCGGA eutK
GAATCGAATGCAACCGGCGC CTGGTCGCTGCAACCGGCGC nuoG
AAGACACGACTTATCGCCAC CAGAGCGAACTTATCGCCAC yqhD
CCCGACCGCTGCGCCTTATC GGATGCGGCTGCGCCTTATC ycjX
AGCATAAAGTGTAAGCCTG AGCATAAAGTGTAAGCCTG rlpA
AAGCATAAAGTGTAAGCCT AAGCATAAAGTGTAAGCCT rlpA
GAAGCATAAAGTGTAAGCC ACCCTTATAAGTGTAAGCC glnD
ATTCCACACAACATACGAGC ATTCCACACAACATACGAGC rlpA
CTATTACGCCAGCTGGCGAA GTCCGTACCCAGCTGGCGAA acrB
CTCTTCGCTATTACGCCAGC AGGGTTATTATTACGCCAGC norV
CTTTCGTTTTATTTGATGCC CTTTCGTTTTATTTGATGCC yccS
CTTTCGTTTTATTTGATGCC CTTTCGTTTTATTTGATGCC yccS
CAAACAACAGATAAAAACGAA CCAGCACCCAGATAAAAACGAA ydcO
CAAACAACAGATAAAAACGAA CCAGCACCCAGATAAAAACGAA ydcO
CGCAGCGGTTCGGGCTGAACG GCTTATTGTCGGGCTGAACG yagG
TATCCGGTAAGCGGCAGGGT GGTGGGCGAAGCGGCAGGGT hslR
TGTGAGTTAGCTCACTCATT TGTGAGTTAGCTCACTCATT yicO
GCTCACTCATTAGGCACCCC GCTCACTCATTAGGCACCCC yicO
GGCTTTACACTTTATGCTTC GTTAATCGACTTTATGCTTC yebQ
CTTCCGGCTCGTATGTTGTG CTTCCGGCTCGTATGTTGTG yicO
AGCGAATAACAATTTACAC ACAGCGGATAACAATTTACACAC yicO
CGTCGTGACTGGGAAAACCC GAAGATAACTGGGAAAACCC hyfH
CACATCCCCCTTTTCGCCAGC CCTTCACCCCTTTTCGCCAGC ybaT
CAGGCATCAAATAAAAACGAA GAAAAAAGAAATAAAAACGAA efeB

RESPONSE: We thank the reviewer for this comment and for pointing out a clarification in our experimental protocol and results which we have added to the revised version (pages 7 and 9). There are three parts to our response.

First, we made no effort to exclude potential mismatches in either the mPool or oPool experiments because these would serve as internal controls (particularly for the oPool) and help us assess the effect of mismatches on measured activity. This is most evident for the mPool where there is a noticeable effect on activity for sgRNA with single mismatches in positions 1-8 (PAM proximal) as compared to single mismatches in other positions. For sgRNA predictions for applications in bacteria, we would avoid designing sgRNAs with any mismatches to the chromosome (as is done in ref 8; Reuter et al. 2021) and have added a statement to that effect on page 7.

Second, not all the sgRNAs identified by the reviewer are in the Illumina sequencing datasets (we assume the reviewer took the list of sgRNAs synthesized for the oPool from Table S3). These sgRNAs that 'dropped out' from the sequencing datasets relative to the input are most likely toxic and cannot be cloned, presumably because they cleave

the *E. coli* chromosome. We overlaid the remaining 33 sgRNAs identified by the reviewer as having potential off-targets in the *E. coli* genome on the activity plots in Figure 2C and 2E (this is now shown in Supplementary Figure S5). These sgRNAs have a wide range of activities. Interestingly, a group of sgRNAs in the upper left quadrant have high activity and very low abundance. We interpret these sgRNAs as potentially toxic. Moreover, 8 of these sgRNAs were among those individually tested in the growth curves shown in Figure 6; 3 of these are toxic. Collectively, this new analysis of the data helps us solidify our initial observation that we can detect sgRNA that are potentially toxic. We thank the reviewer for pointing this out and improving our paper.

Third, the *ccdB* gene on the pTox is regulated by the *lac* promoter, thus some sgRNAs appear to be targeted to the chromosomal *lac* operon. However, both the *E. coli* strains we used (EPI300 and NEB 5-alpha) contain the common $\Delta lacZM15$ mutation which deletes 15 residues of *lacZ* for alpha-complementation. Thus, a subset of the sgRNAs in the oPool experiment (and some of those identified by the reviewer) do not have any exact or near-match targets on the chromosome. The remaining sgRNAs are plotted in Supplementary Figure S5 as discussed above. We have also modified Figure 4A to provide more detail about sgRNA targeting on pTox by identifying regulatory elements.

We thank this reviewer for their fine eye for detail as this has considerably strengthened our results.

3. Is that possible that the differences in growth curves were due to the leaky gene expression?

RESPONSE: We are unclear about what the reviewer means by differences in the growth curves. If the reviewer is referring to the repressed versus induced conditions for each of the sgRNA growth curves, we discussed on page 10 that the repressed growth conditions included glucose in the media to repress SpCas9 expression. Because glucose is a preferred carbon source to arabinose, inclusion in the media would promote more robust growth as evidenced by higher OD values and more rapid growth. If the reviewer is referring to differences between growth curves for different sgRNAs, then leaky expression would only exacerbate any sgRNA-dependent toxicity and be evidenced by lower OD values in the repressed condition. Indeed, we do see several sgRNAs that follow this pattern, and all are classified as toxic in Figures 6E and 6F.

4. When froze the CNN layers, the trainable variables dramatically decreased. Therefore, is that possible that the increased performances were due to the avoidance of the overfitting problem instead of transfer learning? In addition, have the authors tried using resnet blocks in the CNN layers?

RESPONSE: When the CNN layers of our final dual branch model are frozen, only the parameters in the RNN and its subsequent dense layers can be trained. These non-frozen layers are comparable in structure and parameter number to the BGRU models that we tested during model architecture selection. During non-transfer learning tests, we found the BGRU models to perform worse than training on the full dual branch

model. This difference in performance is more pronounced when comparing the non-transfer learning BGRU model results to those obtained when transfer learning with the dual branch model (Supplementary Table S6). Based on the similar likelihood of overfitting for the BGRU models and unfrozen layers of our dual branch model, we believe it is safe to attribute the increase in performance to the use of transfer learning.

We have not yet explored the use of resNET blocks in the CNN layers of our model. Our primary result is the use of transfer learning to improve model performance, a process during which all CNN layers within the model are frozen. We intend to examine the use of resNET blocks, amongst other deep learning methodologies, in our upcoming work.

Minor:

1. In Fig. 1A, the text "growth curve" was obscured by the figure.

RESPONSE: This has been fixed. In our version it was Figure 2A, not Figure 1A, hopefully this was not a glitch introduced during the submission process.

Reviewer #2 (Remarks to the Author):

The paper by Edgell and colleagues deals with the problem of predicting on-target activity of Cas9 sgRNA guides. The authors claim that "current bacterial sgRNA activity models struggle with accurate predictions and do not generalize well", which is true. The lack of generalization in those models is because they are species-specific (mostly designed for humans or other model organisms) and there is not enough data for *E. coli*. Quoting the authors: "the evidence indicates that there is a pressing need for additional high-quality bacterial sgRNA activity data sets to validate and generalize previous findings, and to provide training data for predictive machine learning models". Based on my experience, I agree with that statement. They also point out that "a complicating factor in assessing sgRNA activity in bacteria is that expression of Cas9 (and dCas9) alone can result in cellular toxicity and slow growth", which complicates the determination of what is a good cutter vs. a bad cutter.

The authors propose a method to generate high-quality training data using a two-plasmid positive selection system. The sgRNA activity can be analyzed by deep sequencing of the sgRNA expression cassette.

First, they test the Guo model [ref 28] and the DeepSgRNA model [ref 34] on their data. Their predictive performance of these models was modest (Spearman ~ 0.5). After explaining and analyzing comprehensively their new two-plasmid survival assay to measure sgRNA activity, they introduce a new deep learning model (crisprHAL) that predict sgRNA Cas9 activity in *E. coli*. crisprHAL was trained on Guo's SpCas9 or Guo's eSpCas9 datasets [ref 28] and when tested on their data, Spearman increased from 0.52 to 0.62

Strengths:

- * The two-plasmid survival assay is clever, and it can distinguish highly active sgRNA guides from toxic ones with poor growth, even in repressed conditions
- * The assay was validated extensively
- * The assay's sensitivity was analyzed via a multiplexed sequencing experiment by introducing single and double nucleotide substitutions: sgRNA with no mismatches had higher activity than sgRNAs with one mismatch which in turn had higher activity than sgRNA with two mismatches
- * They showed that growth curves of individual sgRNAs can identify toxic guides: several sgRNAs that would be considered active based on their relative abundance in sequencing experiments had indeed high levels of toxicity
- * Overall the proposed methodology is sound
- * crisprHAL achieves rank correlation of 0.630 when trained on the Guo's dataset and tested on their dataset, better than the Guo model prediction which had a rank correlation of 0.52 (Figure 1)
- * The paper is well written, with high quality figures; there is enough details to reproduce the results

RESPONSE: We thank the reviewer for their positive and constructive comments.

Weaknesses:

- * The dataset produced with this new assay is very very small; 279 guides for TevSpCas9 and 302 for SpCas9. This is completely inadequate to train a deep learning model. The community needs LARGE high-quality sgRNA activity datasets. Can this assay scale to produce much bigger datasets?

RESPONSE: We thank the reviewer for this comment and have included a new Figure 8 that emphasizes the initial training set of ~45,000 sgRNAs from Guo and the transfer learning component of crisprHAL. After transfer learning the rank correlation prediction scores increase from 0.53 to 0.63 for TevSpCas9, showing that transfer learning with small datasets is effective. We note that the assay can be scaled because we can clone large fragments of DNA into pTox to screen pools of sgRNAs in the enrichment experiments. We did this experiment for the revised manuscript, as shown in Figure 8G and 8H, by testing activity of ~300 sgRNAs on the Salmonella *katG* gene that was cloned in pTox.

- * How many parameters does crisprHAL has? Figure 7A shows a complex architecture with many many layers. Considering that Guo's training set has 40-45 thousand guides

(which is not very many for deep learning), how do you know that your model is not over-fitting and just "storing" the guides encoded as weights in the network? if so, the model would not be able to generalize at all -- which is one of your stated objectives

RESPONSE: crisprHAL has approximately 450,000 parameters and the reviewer is correct, with this many parameters memorization is a key concern. However, we validated the model on two independent datasets; the *Salmonella katG* gene, cloned in *E. coli*, and the *Citrobacter rodentium* genome *in situ*. Both datasets have sgRNAs that are completely independent in sequence from the original datasets and indeed the targeted sequences have different codon usages and dinucleotide frequencies. The rank correlations for the new datasets ranged from 0.612-0.678. Thus, crisprHAL is making true predictions and not simply memorizing pre-existing sgRNA sequences. This new data is included in a new section in the results ("crisprHAL predictions are generalizable to other bacteria" on page 16) and Figure 8.

* It would be interesting to compare pros and cons of the two-plasmid survival assay to the functional screens developed for non conventional yeasts in doi:10.1038/s41467-022-28540-0 and doi:10.1016/j.ymben.2019.06.007;

RESPONSE: We thank the reviewer for this suggestion, but as we show below in response to the next criticism, the differences in sgRNA prediction that is evident when comparing eukaryotic and prokaryotic models suggest that these data are incommensurate.

* It would be informative to compare the predictive ability of DeepGuide (doi:10.1038/s41467-022-28540-0) with crisprHAL, but training DeepGuide on the Guo's dataset and testing it on your new dataset

RESPONSE: We have repeated the analysis as requested by the reviewer and included other eukaryotic-specific models for comparison (DeepHF, C-RNNCrispr, DeepSpCas9, and CrisprNET). This data is now presented as Supplementary Figure S1. In brief, we found a Spearman correlation of between -0.2 and 0.1 (indistinguishable from random chance) for the above 4 models. We re-trained DeepGuide on the Guo eSpCas9 dataset (as we did for our base model) and then assessed the predicted versus measured activity for the pTox TevSpCas9 dataset. We found a Spearman rank correlation of 0.505. These results only reinforce the observation that most models do not generalize well outside of the datasets they were trained on (ref 26; Konstantakos et al. 2022) and those datasets that report Cas9/sgRNA activity in eukaryotes are not transferable to bacterial predictions.

* Supplementary Tables S1-S4, which were given to me as hundreds of pages in PDF format, are unreadable and would not be useful to anyone in this format. I strongly suggest distributing them as Excel files.

RESPONSE: We uploaded the files in Excel format. They were likely converted to PDF during the manuscript formatting stage (which we have no control over). We have now

deposited them on FigShare at 10.6084/m9.figshare.23593782 for the review process.

Reviewer #3 (Remarks to the Author):

In this manuscript, the authors claim that they have developed an sgRNA bacterial activity prediction tool. This work is solid with interesting results. However, it looks to me that the progress is incremental for this high-impact journal, and its consideration for publication in Nature Communications would be premature as it. The following comments are made to strengthen the manuscript.

RESPONSE: We thank the reviewer for their comments. As outlined in the specific responses below, we suggest that our work will be transformational.

1. Recently, there have been many papers that report the use of an sgRNA bacterial activity prediction tool or its development. Although they discuss the report by Guo et al., I find that multiple recent papers are missing. For the balanced introduction and discussion sections, I highly recommend that the authors do an extensive literature search and cite relevant papers that refine the existing models or use such tools.

RESPONSE: We are confused as to the papers to which the reviewer is referring. Using the PubMed search terms “sgRNA bacterial activity prediction” results in 11 hits (PMIDs 27661255, 33503261, 29982721, 28146356, 35177617, 36382194, 34508657, 32963084, 33080209, 28985763, 25587897). Only one of these papers (PMID 29982721) describes an sgRNA activity prediction tool (this is the Guo paper). The other hits refer to using bacterial SpCas9 to test activity in eukaryotic systems, or development of webserver to predict activity in a wide range of organisms, or review papers. Thus, we find that the Guo and deepSgRNAbacteria are the only two published models in the primary literature, but agree that there are review and other derivative works. There is more extensive literature on developed sgRNA activity prediction models for eukaryotic organisms, but these are not directly relevant as we showed in the response to reviewer 2 (this is Supplementary Figure S1).

2. To make this work novel and impactful, the generalizability should be better tested and demonstrated. To this end, I highly suggest that they perform new experiments to include more data from different representative species and genera. Without such results, their work looks incremental.

RESPONSE: We thank the reviewer for the suggestion and the push to test the generalizability of the mode and believe this is an extension of the criticism of reviewer #2 that crisprHAL could be ‘memorizing’. To address these significant concerns, we have included new data to address the issue of generalizability, in two ways. First, we cloned the *katG* gene from *Salmonella* and tested a pool of ~300 sgRNAs. As shown in Figure 8, we observed a correlation of 0.68 between measured activity and predicted crisprHAL activity. Second, we cloned a large pool of ~40,000 sgRNAs designed against the *Citrobacter rodentium* genome and conducted a depletion experiment where

active (or toxic) sgRNAs will kill *Citrobacter* and thus become depleted in the sequencing output relative to inactive or weakly active sgRNAs. As shown in Figure 8, we found a correlation of 0.64. In both cases, the model performed well despite targeting completely independent sequences that contain different codon usages and dinucleotide frequencies. It is important to note that this represents the first time that any bacterial sgRNA prediction model has been tested outside of the initial dataset it was generated on, let alone that the model performed so well clearly showing the transformative potential of the methods and the prediction model. Thus, based on our results, crisprHAL clearly generalizes to different organisms and to sequences with different underlying nucleotide compositions.

In addition, our positive selection approach identifies toxic sgRNAs; no other method or assay has been able to do this and the recognition of toxic sgRNA sequences will remove a major confounding variable when designing CRISPR-based screens and tools. Finally, crisprHAL does generalize to datasets generated by other groups, specifically the Guo dataset, and to datasets generated by different assays as outlined above for the *Citrobacter* experiment. This is shown in Figure 8.

3. Toxicity is a very complex phenotype, and the CRISPR toxicity has yet to be elucidated. It is known that there are big differences in Cas/gRNA toxicity between different species or genera. Thus, along with the above comment, the authors should provide more data by testing their systems in more species and genera.

RESPONSE: Please see our response to point #2 above. However, we agree that it will be important to recapitulate the positive selection screen in different model bacterial species. However, this will require completely re-engineering the expression system and identifying appropriate selective gene(s) and this is the focus of our next 5-year research grant application.

4. Many different types of sgRNA prediction tools have been developed, including off-target effect prediction, strain-level specificity prediction, and activity prediction. Given that tools are eventually used for practical applications, I highly recommend that the authors discuss the usage of their 'generalizable' tools from the application perspective

RESPONSE: We have added a section to the discussion regarding the applications of generalizable bacterial sgRNA prediction models.

Reviewers' Comments:

Reviewer #1:

Remarks to the Author:

The authors did not address our legitimate concerns. We believe that their work lacks rigor in two aspects:

First, their experiments lacked proper controls. As we mentioned, some guides can lead to off-target cleavage, which can be lethal for *E. coli*. Since the authors used growth fitness as the screening phenotype, off-target effects can compromise the reliability of their results. For instance, some guides may effectively cut the pTox plasmid, improving growth rate, while also cleaving the genome, which is toxic to *E. coli* growth. Therefore, the authors should have conducted screening experiments without the pTox plasmid to measure the impact of off-target cleavage on growth. Additionally, it's worth noting that we only pointed out some single mismatched sequences in the oPool. Multiple mismatched sequences, particularly those with mismatches in the PAM-distal region, may still have off-target effects. For the mPool, this kind of control is also required.

Second, the authors stated that the dCas9 control is not necessary, yet their training data from Guo et al. included a dCas9 control. Can the authors clarify why the model learned from Guo's data can be effectively applied to their case? Alternatively, can the authors demonstrate the validity and generalizability of their model using other data sources?

Reviewer #2:

Remarks to the Author:

Thanks for the revision. The manuscript has improved considerably.

Reviewer #3:

Remarks to the Author:

The authors have addressed my comments.

Response to reviewer comments for

A generalizable Cas9/sgRNA prediction model using machine transfer learning with small high-quality datasets (NCOMMS-23-12560-T)

Dalton T. Ham^{1,+}, Tyler S. Browne^{1,+}, Pooja N. Banglorewala¹, Tyler L. Wilson², Richard K. Michael², Gregory B. Gloor^{1,*}, David R. Edgell^{1,*}

¹Department of Biochemistry, Schulich School of Medicine and Dentistry, London, ON, N6A5C1, Canada

²Tesseraqt Optimization Inc, Toronto, ON, Canada

+These authors contributed equally to this work

*Corresponding authors: dedgell@uwo.ca, ggloor@uwo.ca

Reviewer Comments

Reviewers #2 and #3 found our changes satisfactory and had no additional concerns.

Reviewer #1 (Remarks to the Author):

The authors did not address our legitimate concerns. We believe that their work lacks rigor in two aspects:

*First, their experiments lacked proper controls. As we mentioned, some guides can lead to off-target cleavage, which can be lethal for *E. coli*. Since the authors used growth fitness as the screening phenotype, off-target effects can compromise the reliability of their results. For instance, some guides may effectively cut the pTox plasmid, improving growth rate, while also cleaving the genome, which is toxic to *E. coli* growth. Therefore, the authors should have conducted screening experiments without the pTox plasmid to measure the impact of off-target cleavage on growth. Additionally, it's worth noting that we only pointed out some single mismatched sequences in the oPool. Multiple mismatched sequences, particularly those with mismatches in the PAM-distal region, may still have off-target effects. For the mPool, this kind of control is also required.*

RESPONSE: We agree with the reviewer that some sgRNAs exhibit toxicity. Indeed, this was one of our major conclusions and our study is one of few that has been able to assess Cas9/sgRNA toxicity in bacteria. However, we disagree that our experiments were uncontrolled. The reviewer is asking us to transform the SpCas9/sgRNA oPool into cells without pTox. This control will not measure on-target SpCas9/sgRNA oPool activity (because none of those sgRNAs were designed to target the *E. coli* chromosome) but would help to identify sgRNAs in the oPool that are toxic, possibly because off-target sites in the chromosome support Cas9/sgRNA-dependent cleavage and killing of *E. coli*. However, as outlined in the response to reviews letter that was submitted with our revised manuscript, we did this control. This data is now presented in Supplementary Figure S5 and Supplementary Table S6. Of the 367 sgRNAs synthesized as the oPool, 36 'dropped-out' and were not found in the experimental

Illumina sequencing datasets and thus are NOT included in the crisprHAL training or test datasets. This drop-out occurred during cloning, that is when we transformed the SpCas9/sgRNA oPool plasmid library into E. coli WITHOUT pTox (as requested by the reviewer). Of the 1140 sgRNAs synthesized in the mPool experiment, we see 1134 in the Illumina sequencing datasets, meaning that 6 sgRNAs ‘dropped out’.

In addition, we are careful to be agnostic as to the mechanism of toxicity. It may be, as the reviewer asserts, due solely to off-target cleavage. However, there may be an as yet undetermined mechanism(s) of toxicity because we observed a slower than expected growth phenotype when some sgRNAs were analyzed individually; some of these sgRNAs have mismatches in the seed region that based on previous literature and our mPool data would preclude activity. We realize that details of the sgRNAs that dropped out from our datasets were not included in the revised manuscript (but were in the rebuttal letter), and these are now in Supplementary Figure S5 and Table S6.

Second, the authors stated that the dCas9 control is not necessary, yet their training data from Guo et al. included a dCas9 control. Can the authors clarify why the model learned from Guo’s data can be effectively applied to their case? Alternatively, can the authors demonstrate the validity and generalizability of their model using other data sources?

RESPONSE: For our experimental system, we believe that dCas9 is not the appropriate control because it has completely different binding characteristics than the non-dead version. However, this does not invalidate prior use of dCas9 for different experimental systems (as performed in the Guo paper). We used the raw dCas9 controlled data to build a dataset with a continuous measurement of sgRNA activity that was compatible with our metric. The dCas9 data was limited by a very low number of replicates (2) and by a very low dynamic range. However, it was extraordinarily useful in providing a broad overview of sgRNA activity for a large number of sgRNAs. We anticipate that the dCas9 data is providing some information on the sequence characteristics needed to produce an active sgRNA that can be refined using the transfer learning approach. However, the improvements to crisprHAL prediction, when we use transfer learning with our newly generated datasets, speak to both the accuracy of our biological data as well as the appropriateness of the crisprHAL machine learning architecture.

We are confused by what the reviewer means by ‘demonstrate the validity and generalizability of their model using other data sources’. At the reviewers’ request, and as described in the response letter and presented in the revised manuscript (Figure 8), we generated two new orthogonal datasets with different (di, tri) nucleotide frequencies from two different species. One dataset tested 31,796 sgRNAs targeted to the *Citrobacter rodentium* chromosome and the second 296 sgRNAs against a cloned fragment of DNA from *Salmonella typhimurium*. These two experiments measure SpCas9 activity in different ways, one is a depletion experiment and the other is an enrichment experiment. In each case, the rank correlation between crisprHAL predicted activity and measured activity is 0.612-0.68. This additional work shows that the model is generalizable to different experimental protocols, different data sources and to

different species, something that has not been shown by any previous ML model for Cas9 activity (bacterial or eukaryotic). We believe that these experiments more than adequately address the comment regarding the 'validity and generalizability' of our model and are unclear what standard is being requested.

Reviewers' Comments:

Reviewer #1:

Remarks to the Author:

The authors have answered all my concerns satisfactorily. This MS. is now acceptable for publication.